# On topological descriptors for graph products

**Mattie Ji**
University of Pennsylvania
mji13@sas.upenn.edu

**Amauri H. Souza**
Federal Institute of Ceará
amauriholanda@ifce.edu.br

**Vikas Garg**
Aalto University
YaiYai Ltd
vgarg@csail.mit.edu

## Abstract

Topological descriptors have been increasingly utilized for capturing multiscale structural information in relational data. In this work, we consider various filtrations on the (box) product of graphs and the effect on their outputs on the topological descriptors - the Euler characteristic (EC) and persistent homology (PH). In particular, we establish a complete characterization of the expressive power of EC on general color-based filtrations. We also show that the PH descriptors of (virtual) graph products contain strictly more information than the computation on individual graphs, whereas EC does not. Additionally, we provide algorithms to compute the PH diagrams of the product of vertex- and edge-level filtrations on the graph product. We also substantiate our theoretical analysis with empirical investigations on runtime analysis, expressivity, and graph classification performance. Overall, this work paves way for powerful graph persistent descriptors via product filtrations. Code is available at https://github.com/Aalto-QuML/tda_graph_product.

## 1 Introduction

Message-passing GNNs are prominent models for graph representation learning [19]. However, they are bounded in expressivity by the WL hierarchy [33–35, 50] and cannot compute fundamental graph properties such as cycles or connected components [10, 18]. Topological descriptors (TDs) such as those based on persistent homology (PH) and the Euler characteristics (EC) can provide such information and thus are being increasingly employed to augment GNNs, boosting their empirical performance [7–9, 24, 48, 51].

In this work, we study two prominent types of topological descriptors based on PH [1, 13] and EC [40, 42, 46] respectively. Both descriptors are defined with respect to a fixed filtration of a graph. On a high level, PH keeps track of the birth and death times of topological features (e.g., connected components and loops) in a (parameterized) graph filtration. EC is a simpler, less expensive invariant that keeps track of the number of vertices minus the number of edges in the filtration. Both PH and EC have found use across a wide range of applications in GNNs [16, 36, 42].

A precise characterization of PH has been established for graphs based on color-based filtrations in [26]. Here, we offer a complete characterization of the expressivity of EC in terms of combinatorial data on the colors of the graph. EC also compares favorably with other TDs in terms of computation, which is another important aspect we explore here. In particular, we streamline PH and EC with what we call their "max" versions. We show that the vertex death times, as well as the cycle birth times, of vertex-level and edge-level max PH are the same. We also establish that max EC retains the expressivity of EC, but is even more-cost effective than EC. Interestingly, we show that this expressivity equivalence on graphs extends to a more general equivalence for color filtrations on finite higher-dimensional simplicial complexes. This immediately opens avenues for integrating max EC as an economical alternative to PH-based TDs into Topological Neural Networks, which generalize GNNs through higher-order message-passing [5, 6, 15, 37, 38] and have recently been shown to benefit from TDs both theoretically and empirically [47].

39th Conference on Neural Information Processing Systems (NeurIPS 2025).

Table 1: Overview of our theoretical results.

| | |
|---|---|
| **Expressive Power of PH and EC (Section 2, 3)** | |
| Matching Vertex Deaths and Cycle Births within max PH | Proposition 1 |
| EC Diagram $\cong$ max EC Diagram on graphs | Theorem 1 |
| The Combinatorial Characterization of EC Diagrams on Graphs | Theorem 2 |
| Generalization EC $\cong$ max EC to Simplicial Complexes | Theorem 3 |
| **Topological Descriptors on Graph Products (Section 4):** | |
| EC of Product Graphs Does Not Add New Information | Proposition 5 |
| PH of (Virtual) Product Graphs $\succ$ PH of Components | Corollary 1 |
| **Computational Methods for Products Filtrations (Section 5):** | |
| Product of Vertex Filtrations $\simeq$ Some Vertex Coloring on Product | Proposition 6 |
| Product of Edge Filtrations $\simeq$ Some Edge Coloring on Product | Proposition 7 |
| Algorithm for 0-th Persistence Pairs for Vertex Product Filtrations | Theorem 4 |
| Algorithm for 0-th Persistence Pairs for Edge Product Filtrations | Theorem 5 |
| Algorithm for 1-st Persistence Pairs for Product Filtrations | Proposition 8 |

From a practical micro-level perspective, the design of effective filtration functions is also paramount. However, almost invariably, filtration functions are either fixed *a priori* or learned without any structural considerations (e.g., symmetries), obfuscating how they are about the corresponding diagrams. We seek to address this gap from a rigorous theoretical perspective, in pursuit of principled design of richer topological descriptors.

The (box) product of graphs [27] naturally leads to symmetries and thus provides an effective starting point for us, where we treat individual components as graph fractals. From a machine learning perspective, graph products also arise naturally in the context of subgraph GNNs [3, 4], serve as a natural tool for modeling and analyzing multi-way data [29, 45] and relational databases (see, e.g., Motivation 1 in Section 4), and have been incorporated in various GNN architectures [14, 43].

For two colored graphs $G$ and $H$, there is a natural coloring on the graph product $G \square H$. This enables us to consider vertex-level and edge-level color-based filtration on $G \square H$. Under a technical modification to the graph product, we show that the PH of the graph product is strictly more expressive than the PH of the individual components. Surprisingly, the benefits of graph products under PH do not extend to EC. Leveraging our complete characterization of the expressive power of EC, we show that taking the EC of graph products provides does not provide any expressivity benefit beyond the EC of individual components.

A notable subcollection of filtrations on graph products is given by the (box) product of filtrations on the individual components. We initiate here a formal analysis by characterizing product filtrations: we establish that the product of vertex filtrations is the max of vertex colors, and that the product of edge filtrations is a certain edge coloring on the product. We also provide algorithms to track the evolution of persistence diagrams under the product filtration induced by vertex filtrations and edge filtrations respectively (in fact, the algorithm for 1-dimensional features works for any product filtrations). This gives a computationally cost-effective method to compute these PH diagrams.

In sum, our work introduces a calculus for topological descriptors under the graph product. We summarize our theoretical results in Table 1, and relegate all the proofs to the Appendix.

## 2 Preliminaries

A **graph** is the data $G = (V, E, c, X)$ where $V$ is a finite vertex set, $E \subseteq V \times V$ is the set of edges, $c : V \to X$ is a vertex coloring function, and $X$ is the finite set of possible colors. An **isomorphism** between graphs $G = (V, E, c, X)$ and $G' = (V', E', c', X')$ is a bijection $h : V \to V'$ such that (a) $c = c' \circ h$ and (b) $(v, w) \in E$ if and only if $(h(v), h(w)) \in E'$. We will only consider finite simple graphs in this work, and all graphs in this work will share the same coloring set.

In this work, we consider two types of color-based filtrations based on vertex-level and edge-level.

**Definition 1** (Color-Based Filtrations). *On a color set $X$, we consider the pair of functions $(f_v : X \to \mathbb{R}, f_e : X \times X \to \mathbb{R}_{>0})$ where $f_e$ is symmetric (ie. $f_e(c, c') = f_e(c', c)$). On a graph $G$ with a vertex color set $X$, $(f_v, f_e)$ induces the following pair of functions $(F_v : V \cup E \to \mathbb{R}, F_e : V \cup E \to \mathbb{R}_{\geq 0})$.*

> 1. *For all $v \in V(G)$, $F_v(v) := f_v(c(v))$. For all $e \in E(G)$ with vertices $v_1, v_2$, $F_v(e) = \max\{F_v(v_1), F_v(v_2)\}$. Intuitively, we are assigning the edge $e$ with the color $c(\arg\max_{v_i} F_v(v_i))$ (the vertex color that has a higher value under $f_v$).*
> 2. *For all $v \in V(G)$, $F_e(v) = 0$. For all $e \in e(G)$ with vertices $v_1, v_2$, $F_e(e) := f_e(c(v_1), c(v_2))$. Intuitively, we are assigning the edge $e$ with the color $(c(v_1), c(v_2))$.*

*For each $t \in \mathbb{R}$, we write $G_t^{f_v} := F_v^{-1}((-\infty, t])$ and $G_t^{f_e} := F_e^{-1}((-\infty, t])$. When there is a clear fixed choice of filtration, we may drop the superscripts and simply write $G_t$.*

Note we used $G_t^{f_v}$ as opposed to $G_t^{F_v}$ to emphasize that the function $G \mapsto (\{G_t^{f_v}\}_{t \in \mathbb{R}}, \{G_t^{f_e}\}_{\mathbb{R}})$ is well-defined for any graph $G$ with the coloring set $X$. The lists $\{G_t^{f_v}\}_{t \in \mathbb{R}}$ and $\{G_t^{f_e}\}_{t \in \mathbb{R}}$ define a **vertex filtration** of $G$ by $F_v$ and an **edge filtration** of $G$ by $F_e$ respectively. It is clear that $G_t^{f_v}$ can only change when $t$ crosses a critical value in $\{f_v(c) : c \in X\}$, and $G_t^{f_e}$ can only change when $t$ crosses a critical value in $\{f_e(c_1, c_2) : (c_1, c_2) \in X \times X\}$. Hence, we can reduce both filtrations to finite filtrations at those critical values.

We now review the two relevant classes of topological descriptors (TDs) for us, namely, persistent homology (PH) and the Euler characteristic (EC).

## 2.1 Persistent Homology

A vertex $v$ (ie. $0$-dimensional persistence information) is born when it appears in a given filtration of the graph. When we merge two connected components represented by two vertices $v$ and $w$, we use a decision rule to kill off one of the vertices and mark the remaining vertex to represent the new connected component. A cycle (ie. $1$-dimensional persistence information) is born when it appears in a given filtration of a diagram, and it will never die. For color-based vertex and edge filtration, there is a canonical way to calculate the **persistence pairs** of a graph with a given filtration. We refer the reader to Appendix A of Immonen et al. [26] for a precise introduction. We say a $0$-th dimensional persistence pair $(b, d)$ is a **real hole** if $d = \infty$, is an **almost hole** if $b \neq d < \infty$, and is a **trivial hole** if $b = d$. We say that a birth or death of a vertex is **trivial** if its persistence tuple is a trivial hole.

**Definition 2.** *Let $f = (f_v, f_e)$ be on the coloring set $X$. The persistent homology diagram of a graph $G$ is a collection $\mathrm{PH}(G, f)$ composed of two lists $\mathrm{PH}(G, f)^V, \mathrm{PH}(G, f)^E$ where $\mathrm{PH}(G, f)^V$ are all the persistent pairs in the vertex filtration $\{G_t^{f_v}\}_{t \in \mathbb{R}}$ and $\mathrm{PH}(G, f)^E$ are all the persistent pairs in the edge filtration $\{G_t^{f_e}\}_{t \in \mathbb{R}}$.*

PH is an isomorphism invariant (Theorem 2 of Ballester and Rieck [2]) that generalizes an older invariant - the homology. We will sometimes use $\beta_0(G)$ and $\beta_1(G)$ to denote the $0$-th and $1$-st Betti number of a graph (ie. the rank of the $0$-th and $1$-st homology).

## 2.2 Euler Characteristic

Including topological features in graph representation learning using persistent homology can be computationally expensive. The Euler characteristic (EC) provides a weaker isomorphism invariant that is easier to compute (but typically less expressive). For a graph $G$, the Euler characteristic $\chi(G)$ is given by $\chi(G) = \#V(G) - \#E(G)$. Given a filtration of a graph $G$, we can track its Euler characteristic throughout the filtration.

**Definition 3.** *Let $f = (f_v, f_e)$ be on the coloring set $X$. Write $a_1 < ... < a_n$ as the list of values $f_v$ can produce, and $b_1 < ... < b_m$ as the list of values $f_e$ can produce. The **EC diagram** of a graph $G$ is two lists $\mathrm{EC}(G, f) = \mathrm{EC}(G, f)^V \sqcup \mathrm{EC}(G, f)^E$, where $\mathrm{EC}(G, f)^V$ is the list $\{\chi(G_{a_i}^{f_v})\}_{i=1}^n$ and $\mathrm{EC}(G, f)^E$ is the list $\{\chi(G_{b_i}^{f_e})\}_{i=1}^m$.*

We would also like to consider a weaker variant of EC that we will call the **max EC**.

**Definition 4.** *Given only a vertex coloring function $f_v : X \to \mathbb{R}_{>0}$. We define the **max EC** **diagram** of $G$ as $\mathrm{EC}^m(G, f_v) := \mathrm{EC}(G, (f_v, h(f_v)))$, where $h(f_v) : X \times X \to \mathbb{R}_{>0}$ is defined*

*as $h(f_v)(c_1, c_2) := \max(f_v(c_1), f_v(c_2))$. Note that max EC is the restriction of EC to only the max-induced edge filtrations.*

The reader might wonder what happens when we could consider a suitable analog of **max PH** as well. We discuss this possibility in Appendix B and give an explicit method to compute it.

**Proposition 1.** *Write $f = (f_v > 0, h(f_v))$, where $h$ is defined in Definition 4. The vertex death times (as a multi-set) of $\mathrm{PH}(G, f)^E$ and $\mathrm{PH}(G, f)^V$ are the same. The cycle birth times are also the same.*

# 3 The Expressive Power of the Euler Characteristics

In this section, we will discuss a surprising equivalence between the expressivity of max EC diagrams and EC diagrams on graphs.

> **Theorem 1.** EC *and* $\mathrm{EC}^m$ *have the same expressive power, that is,* EC *can differentiate non-isomorphic graphs $G$ and $H$ if and only if* $\mathrm{EC}^m$ *can also differentiate them.*

In fact, Theorem 1 will follow as an immediate corollary of a stronger theorem that we will prove below, where we give a complete characterization of the expressivity of (max) EC diagrams on graphs. We will first discuss two complete characterizations of the expressivity of EC with respect to vertex-level and edge-level filtrations.

Let $G, H$ be two graphs both on $n$ vertices. We will consider the following combinatorial objects.

**Notation:** We use $E_G(a, b)$ to denote the set of edges in $G$ with endpoints being a vertex of color $a$ and a vertex of color $b$, $V_G(a)$ to denote the set of vertices in $G$ with color $a$, and $G(c), H(c)$ to denote the subgraphs of $G, H$ generated by the vertices of color $c$.

In our next results, Proposition 2 and Proposition 3, we establish that the (max) EC diagrams of a graph $G$ may be completely interpreted in terms of the combinatorial data provided by the objects $E_G(a, b), V_G(a)$ and $G(c)$ defined in the notations above.

**Proposition 2** (Characterization of (max) EC Edge Filtrations)**.** *The following are equivalent:*

1. *For all (symmetric) edge color functions $g : X \times X \to \mathbb{R}$, $\mathrm{EC}(G, g)^E = \mathrm{EC}(H, g)^E$.*
2. *For all vertex color functions $f : X \to \mathbb{R}$, $\mathrm{EC}^m(G, f)^E = \mathrm{EC}^m(H, f)^E$.*
3. *$\#E_G(a, b) = \#E_H(a, b)$ for all colors $a, b \in X$.*

**Proposition 3** (Characterization of (max) EC Vertex Filtrations)**.** *The following are equivalent:*

1. *For all vertex color functions $f : X \to \mathbb{R}$, $\mathrm{EC}(G, f)^V = \mathrm{EC}(H, f)^V$.*
2. *For all vertex color functions $f : X \to \mathbb{R}$, $\mathrm{EC}^m(G, f)^V = \mathrm{EC}^m(H, f)^V$.*
3. *For all $a \neq b, c \in X$, $\chi(G(c)) = \chi(H(c))$ and $\#E_G(a, b) = \#E_H(a, b)$.*

Combining the statements of Proposition 2 and Proposition 3, we obtain the following theorem.

> **Theorem 2** (Characterization of (max) EC Diagrams)**.** *The following are equivalent:*
> 1. *$G$ and $H$ have the same* EC *diagram for any choice of coloring functions.*
> 2. *$G$ and $H$ have the same max* EC *diagram for any choice of coloring functions.*
> 3. *$\#V_G(c) = \#V_H(c)$ for all $c \in X$ and $\#E_G(a, b) = \#E_H(a, b)$ for any $a, b \in X$.*

While the majority of our work is focused on graphs, the equivalence of EC and max EC is a special case of a more general equivalence on the level of (colored) simplicial complexes (see Appendix D).

> **Theorem 3.** *There is a natural extension of the definition of EC and max EC to colored simplicial complexes such that they have the same expressive power for color-based filtrations.*

# 4 Persistent Topological Descriptor for Graph Products

When dealing with large or complex graphs that have the structural property of being some product, it is often easier to work with the components of the graph product rather than the graph as a whole.

There are common situations where graph data have a natural product structure (e.g. relational databases [41], Hamming graphs [23, 25]) that can be analyzed using the methods here.

Our focus is to look at topological descriptors on product of graphs, as a special kind of structure.

**Definition 5.** *Let $G, H$ be graphs, the **box product (Cartesian product)** of $G$ and $H$ is the graph $G \Box H$, where the vertex set of $G \Box H$ is the set $\{(g, h) \mid g \in V(G), h \in V(H)\}$ and the edge set is constructed as follows. For vertices $(g_1, h_1)$ and $(g_2, h_2)$, we draw an edge if (1) $g_1 = g_2$ and $h_1 \sim h_2$ in $H$ or (2) $h_1 = h_2$ and $g_1 \sim g_2$ in $G$. Here, by $h_1 \sim h_2$, we mean that $h_1$ and $h_2$ are related by an edge in $H$ (and similarly for $g_1 \sim g_2$). Note that if one of $G$ and $H$ is empty, then the box product is empty.*

*There is a natural coloring on $G \Box H$ where a vertex $(g, h) \in V(G \Box H)$ is assigned the color $(c(g), c(h))$. For simplicity, we will view $(c_1, c_2)$ and $(c_2, c_1)$ as the same color for all $c_1, c_2 \in X$. Unless mentioned otherwise, a vertex-level (resp. edge-level) color-based filtration on $G \Box H$ is always taken to be with respect to the product color structure above.*

**Motivation 1.** Graph products provide a natural model to convert database analysis questions to inputs for GNNs. For example, suppose one has two tables - *Table A* is the wage income of residents in New York, and *Table B* is the wage income of residents in London, with some additional demographic / financial information for both. For each table, we can view the rows as nodes and assign a similarity metric on the rows such that two nodes with sufficient similarity are linked with an edge. Suppose our goal is to compare the residents in New York against the residents in London, then one natural operation would be to take the Cartesian product of the tables, call this new table *Table C*. A reasonable graph one can make out of *Table C* is as follows - the nodes are still the rows. For person $X$ in *Table A* paired with person $Y$ in *Table B*, in order to analyze how $X$ compares with $Y$ and those similar to $Y$, we ought to look at how $X$ connects with everyone in *Table B* that are connected to Y under the similarity metric before. What this means in terms of graphs is that we should link the node $(X, Y)$ to $(X, Y')$ for all $Y'$ that is linked to $Y$. We can also do this symmetrically, which produces exactly the box product of *Table A* and *Table B*.

We can define PH and EC on graph products with respect to the color-based filtrations in Definition 5. We wish to examine whether the graph product contains information that the components do not give. Given two graphs $G$ and $H$ and suppose for contradiction that they are isomorphic, then it follows that their products $G \Box G, G \Box H$, and $H \Box H$ are all isomorphic graphs. If we could show that any two product graphs in the triplet are not isomorphic, then $G$ and $H$ are not isomorphic.

In the case of PH, the graph product does capture structure not seen by the individual components.

**Proposition 4.** *There exist graphs $G$ and $H$ such that PH cannot tell apart, but PH can tell apart $G \Box G$ and $H \Box H$. Furthermore, an example is given in Figure 1.*

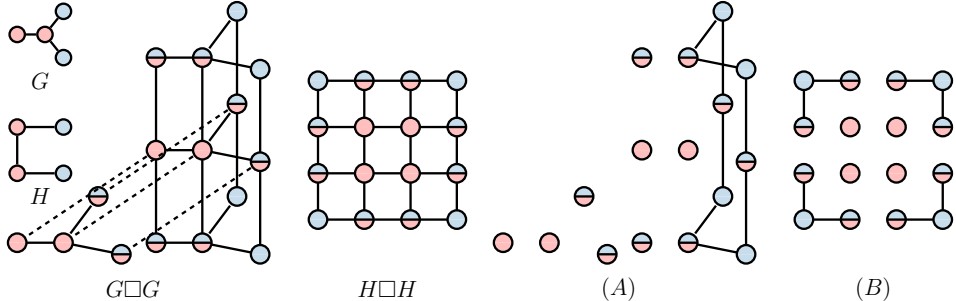

Figure 1: Graphs $G$ and $H$ that vertex-level and edge-level PH cannot tell apart, but they can be differentiated by running edge-level PH on $G \Box G$ and $H \Box H$. Here, connecting the edges between vertices of color (blue, blue) and (red, blue) creates a non-trivial cycle in $(A)$ and no cycles in $(B)$.

The EC, on the other hand, does not obtain any more expressivity over the product filtration.

**Proposition 5.** *Let $G$ and $H$ be graphs with the same number of vertices. If EC cannot differentiate $G$ and $H$, the EC diagrams for $G \Box G$, $G \Box H$, and $H \Box H$ are also all the same.*

The proof of Proposition 5 comes from a direct application of Theorem 2, as we will show in the Appendix that $G \Box G, H \Box H$, and $G \Box H$ all have the same combinatorial data in the statement.

In light of Proposition 4, a natural question to ask is - *can the PH of the product contain **strictly more** information than the PH of the two components?* The answer is guaranteed to be yes if we adjoin a "virtual node" in the following sense.

**Definition 6.** *Let $G = (V, E, c, X)$ be a graph, we define $G_+$ as $(V \sqcup \{*\}, E, c', X \sqcup \{virtual\})$ where $*$ is a disjoint basepoint and we define $c'(*) = virtual$ and $c'(x) = c(x)$ for all $x \in X$. For any color-based filtration of $G_+$, we also require the vertex $*$ to be born at time $-\infty$ (ie. it appears before any other vertices).*

With the example given in Proposition 4, we have the following corollary.

**Corollary 1.** *Let $G$ and $H$ be graphs, then the PH of $G_+ \square G_+, G_+ \square H_+, H_+ \square H_+$ contain strictly more information then PH of $G$ and $H$.*

In other words, taking the PH of "virtual" graph products contain strictly more information than comparing the PH of each component. We remark that the EC of "virtual" graph product still would not add more information. This is because if graphs $A$ and $B$ have the same EC diagram for all possible filtrations, so does $A_+$ and $B_+$, allowing us to apply Proposition 5.

## 5 Product Filtrations

In this section, we focus on a strict subset of possible filtrations on the graph product. Given graphs $G$ and $H$, we investigate what information about the graph product $G \square H$ can be recovered from filtrations on $G$ and $H$ alone. Formally, we want to consider filtrations of the following form.

**Definition 7.** *Let $G$ and $H$ be graphs with filtration functions $f_G$ and $f_H$ respectively. We can define a **product filtration** of $G \square H$ with respect to the two filtration functions as the following - let $t \in \mathbb{R}$, then the subgraph $(G \square H)_t$ is exactly $G_t^{f_G} \square H_t^{f_H}$.*

### 5.1 Characterization of Vertex-level and Edge-level Product Filtrations

The two notable forms of filtrations on $G$ and $H$ we want to consider are vertex-level and edge-level filtrations in the sense of Definition 1. Here, we give a characterization of their respective product filtrations in terms a convenient coloring on $G \square H$.

**Proposition 6.** *Suppose $f_G$ and $f_H$ are injective vertex color functions. The product filtration with respect to $f_G$ and $f_H$ is equivalent to the filtration on $G \square H$ given by the vertex coloring function $F$, where $F((g, h)) = \max(f_G(c(g)), f_H(c(h)))$, for all $(g, h) \in V(G \square H)$. In particular, this implies that the persistence diagrams of this product filtration are well-defined. We write $F = f_G \square f_H$ when the context is clear.*

**Proposition 7.** *Suppose $f_G > 0$ and $f_H > 0$ are injective edge color functions. The product filtration with respect to $f_G$ and $f_H$ is equivalent to the filtration on $G \square H$ given by the function $F$ where where $F$ sends all vertices to 0 and for all $e = ((g_1, h_1), (g_2, h_2)) \in E(G \square H)$, $F(e) = f_H(c(h_1), c(h_2))$ if $g_1 = g_2$ and $F(e) = f_G(c(g_1), c(g_2))$ if $h_1 = h_2$. Note that $g_1 = g_2$ and $h_1 = h_2$ cannot both be satisfied on a simple graph. We write $F = f_G \square f_H$ when the context is clear.*

### 5.2 Computational Methods for Product Filtrations

In this section, we discuss algorithmic procedures to compute the PH of the vertex-level and edge-level persistence diagrams. The computational methods we obtain for 1-dimensional persistence diagrams is more general, so we will discuss them separately. Note that although we used color-based filtrations in the expressivity analysis above, the algorithms devised here applies to any pair of filtrations. For a graph $G$, we let $m_G$ and $n_G$ be the number of vertices and edges respectively.

Following Proposition 6, we can in fact obtain an algorithmic procedure to keep track of how the 0-th dimensional persistence diagram of $G \square H$ changes under the product of vertex filtrations.

> **Theorem 4.** *Let $(G, f_G), (H, f_H)$ be two vertex-based filtrations, then the 0-dimensional PH diagram for $(G \square H, f_G \square f_H)$ can be computed using Algorithm 1. This can be equivalently be computed by Algorithm 3 in the Appendix. The runtime is $O(n_G n_H + m_G \log m_G + m_H \log m_H + n_G + n_H)$, assuming the coloring set has constant size.*

We remark that the runtime complexity in Theorem 4 is more efficient than the naive method. A naive calculation for vertex-level product filtrations would be to run PH on the entire graph product directly. This incurs a runtime of $O(n_G n_H + (n_G m_H + n_H m_G)log(n_G m_H + n_H m_G))$. This is because, by the discussions before 4.1 of [24], the PH of a connected graph with $r$ edges can be calculated in $O(r \log(r))$ time. If the graph is not connected, the runtime is $O(s + r \log(r))$, where $s$ is the number of vertices, to account for the worst-case scenario where the graph is totally disconnected. We demonstrate the runtime benefits of Theorem 4 empirically later in Figure 3.

We now describe a concrete example of computation using Algorithm 1 outlined in Theorem 4.

**Example 1.** *Consider the graphs $G = H$ in Figure 2, and let $a_0 = f_v(\text{blue}) < a_1 = f_v(\text{red})$. Here we take $f_G = f_H = f_v$. Observe that both $G$ and $H$'s individual filtrations proceed by spawning two blue vertices at $a_0$ and completing to the entire graph at $a_1$. We would like to compute the 0-dimensional PH of the product of vertex filtrations. Before the for-loop, we initialize 16 birth-death pairs with 4 copies of $(a_0, -)$ and 12 copies of $(a_1, -)$ (we omit their vertex representatives for simplicity).*

*We start with $t = a_0$. Note that since we removed $-\infty$ before the loop, the index 1 elements in the precomputed lists correspond to the relevant values at time $a_0$. Since there are no deaths in either $G$ or $H$ at $a_0$, we skip the two internal for-loops. $\text{bett}_H[0]$ is empty (since this is recorded at $-\infty$), and $\text{betti}_G[1] \times \text{births}_H[1]$ represent the 4 blue vertices and there are no non-trivial deaths. We then mark all tuples that are not the 4 blue vertices with birth time $a_0$ to die at $a_0$, but there are no such tuples. Thus, we conclude that nothing happens at $t = a_0$.*

*At $t = a_1$, 2 red vertices and 1 blue vertex die for both $G$ and $H$. Only the blue vertex death is non-trivial, so in the first inner for-loop we mark 2 of the blue vertices in $G \square H$ to die as $(a_0, a_1)$. In the second inner for-loop, one of the blue vertex is marked already, and we mark another tuple to $(a_0, a_1)$. This produces 3 copies of $(a_0, a_1)$. There are no non-trivial births at this stage, so we mark all vertices born at $a_1$ to die at $a_1$, which makes 12 copies of $(a_1, a_1)$.*

*At $t = +\infty$, the final blue vertex dies for both $G$ and $H$, so we mark the final vertex in $G \square H$ to die as $(a_0, \infty)$. There are no non-trivial deaths, and also no remaining tuples to fill out, so we stop.*

---

**Algorithm 1** Computing the 0-dim PH Diagrams in Product of Vertex-Level Filtrations

---

**Input:** The vertex-level filtrations $(G, f_G)$ and $(H, f_H)$
**Output:** The 0-dim PH diagram for $(G \square H, f_G \square f_H)$
1: $\text{impl}_G, \text{impl}_H \leftarrow$ 0-dim PH diagrams for $(G, f_G)$ and $(H, f_H)$.
2: filtration_steps $\leftarrow \{-\infty\} \cup$ filtration steps for $f_G$ and $f_H \cup \{+\infty\}$, and ordered.
3: $\text{births}_G$, $\text{births}_H \leftarrow$ list of tuples born non-trivially in $G$ (resp. $H$) at each time in filtration_steps.
4: $\text{betti}_G, \text{betti}_H \leftarrow$ list of vertices alive representing components of the subgraph of $G$ (resp. $H$) at each time in filtration_steps.
5: $\text{deaths}_G, \text{deaths}_H \leftarrow$ list of tuples that die non-trivially in $G$ (resp. $H$) at each time in filtration_steps.
6: Remove $\{-\infty\}$ from filtration_steps.
7: **output** $\leftarrow [(i, j, \max(f_G(i), f_H(i)), \text{None})$ **for** $(i, j)$ in $V(G) \times V(H)]$.
8: **for** i in [0, len(filtration_steps)] **do**
9:     $a_i \leftarrow$ filtration_steps$[i]$.
10:     **for** v in $\text{deaths}_H[i+1]$ **do**                ▷ Marking Non-Trivial Deaths
11:         Mark all tuples $p$ in **output** with $p[1] = v, p[2] < a_i, p[3] = \text{None}$ with $p[3] \leftarrow a_i$.
12:     **for** w in $\text{deaths}_G[i+1]$ **do**               ▷ Marking Non-Trivial Deaths
13:         Mark all tuples $p$ in **output** with $p[0] = w, p[2] < a_i, p[3] = \text{None}$ with $p[3] \leftarrow a_i$.
14:     nt_births $\leftarrow \text{births}_G[i+1] \times \text{betti}_H[i] + \text{betti}_G[i+1] \times \text{births}_H[i+1]$.
15:     nt_deaths $\leftarrow \text{births}_G[i+1] \times \text{deaths}_H[i+1]$.
16:     nt_births $\leftarrow$ nt_births $-$ nt_deaths.
17:     Mark all tuples $p$ in **output** with $(p[0], p[1]) \notin$ nt_births, $p[2] = a_i, p[3] = \text{None}$ with $p[3] \leftarrow a_i$.         ▷ Marking Trivial Deaths
18: **return** $[(p[2], p[3])$ **for** p in **output**]

---

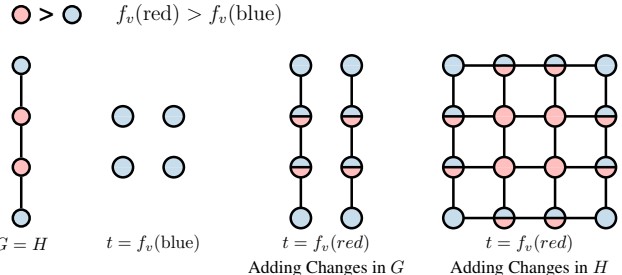

Figure 2: The product of the vertex filtrations on $G = H$ given by $f_v(\text{red}) > f_v(\text{blue})$. See Example 1 for how the algorithm in Theorem 4 is used to compute the PH of this filtration.

Now we describe an algorithm for the 0-th dimensional persistence diagram of the product of edge filtration. Since all vertices are born at the same time in this filtration, it suffices for us to keep track of the number of deaths at each time.

**Theorem 5.** *Let $(G, f_G), (H, f_H)$ be two edge-based filtrations. The 0-dimensional PH diagram for $(G\square H, f_G\square f_H)$ can be computed using Algorithm 2. The runtime is $O(\max(n_G \log n_G + n_H \log n_H, m_G \log m_G + n_G, m_H \log m_H + n_H))$.*

The runtime in Theorem 5 is again an improvement over the naive case, which is $O(n_G n_H + (n_G m_H + n_H m_G) log(n_G m_H + n_H m_G))$, and we empirically assess their performance in Figure 3. We give a computational example using Theorem 5 in Example 2 of the Appendix. Finally, we describe a procedure to calculate the 1-dimensional persistence diagrams for a general product filtration.

---

**Algorithm 2** Computing the 0-dim PH Diagrams in Product of Edge-Level Filtrations

---

**Input:** The edge-level filtrations $(G, f_G \geq 0)$ and $(H, f_H \geq 0)$
**Output:** The 0-dim PH diagram for $(G\square H, f_G\square f_H)$
1: $\text{impl}_G, \text{impl}_H \leftarrow$ 0-dim PH diagrams for $(G, f_G)$ and $(H, f_H)$.
2: filtration_steps $\leftarrow \{-\infty\}\cup$ filtration steps for $f_G$ and $f_H \cup\{+\infty\}$, and ordered.
3: $\text{deaths}_G$, $\text{deaths}_H \leftarrow$ list of number of vertices that die at each time in filtration_steps.
4: $\text{still\_alive}_G$, $\text{still\_alive}_H \leftarrow$ list of number of vertices that are still alive at each time in filtration_steps.
5: Remove $\{-\infty\}$ from filtration_steps.
6: **output = []**
7: **for** i in [0, len(filtration_steps)] **do**
8:     $a_i \leftarrow$ filtration_steps$[i]$.
9:     num $\leftarrow \text{still\_alive}_H[i]\cdot\text{deaths}_G[i+1] + \text{still\_alive}_G[i+1]\cdot\text{deaths}_H[i+1]$.
10:     **output +=** $[(0, a_i)$ for i in range(0, num)]
11: **return output**

---

**Proposition 8.** *Let $\{G_t\}$ and $\{H_t\}$ be two filtrations of $G$ and $H$ respectively. Consider the induced filtration on $G\square H$ given by $(G\square H)_t := G_t\square H_t$, and let $a_1 < a_2 < ... < a_n$ mark the times where $(G\square H)_t$ changes. The 1-dimensional PH of this filtration is exactly the data of $\beta_1(G_{a_1}\square H_{a_1})$ copies of $(a_1, \infty)$ and $\beta_1(G_{a_i}\square H_{a_i}) - \beta_1(G_{a_{i-1}}\square H_{a_{i-1}})$ copies of $(a_i, \infty)$ for $i > 1$. The runtime is $O(m_G \log m_G + n_G + m_H \log m_H + n_H)$ because, for any two graphs $A$ and $B$, we may compute $\beta_1(A\square B) = \#E(A)\#E(B) + \beta_0(A)\beta_0(B) - \chi(A)\chi(B)$.*

**Limitation.** The algorithms presented here only cover a subcollection of possible color-based filtrations on the graph product $G\square H$. In general, for a filtration on $G\square H$ that does not arise from the product of filtration on $G$ and $H$, there is no reason to expect the PH for the filtration on $G\square H$ can be neatly decomposed similarly to the algorithms in this section. Computing PH on graph products with color-based filtrations outside of product filtrations may be more computationally expensive.

Table 2: Accuracy Results: Standard PH vs. PH on graph products (GProd) on pairs of Cayley Graphs and three datasets from the BREC benchmark. We use filtrations induced by three functions: edge closeness, Jaccard index, and square clustering coefficient. Overall, persistent topological descriptors computed on graph products yield improved performance in distinguishing non-isomorphic graphs.[1]

| Filtration | Method | Minimal Cayley Graphs | | | | | | | BREC Datasets | | |
| | | c-12 | c-16 | c-20 | c-24 | c-32 | c-36 | c-63 | Basic (60) | Reg. (50) | Ext. (100) |
|---|---|---|---|---|---|---|---|---|---|---|---|
| Closeness | $PH^E$ | 100.0 | 100.0 | 100.0 | 99.85 | 99.93 | 99.93 | 100.0 | 93.33 | 86.00 | 71.00 |
| | $GProd^E$ | 100.0 | 100.0 | 100.0 | **100.0** | 99.93 | **100.0** | 100.0 | **100.0** | **100.0** | **94.00** |
| Jaccard Index | $PH^E$ | 95.24 | 83.33 | 60.71 | 85.59 | 76.50 | 84.33 | 73.33 | 98.33 | 94.00 | 56.00 |
| | $GProd^E$ | 95.24 | 83.33 | 60.71 | 85.59 | 76.50 | 84.33 | 73.33 | 98.33 | 94.00 | **61.00** |
| Clustering | $PH^V$ | 100.0 | 100.0 | 100.0 | 95.80 | 97.64 | 96.52 | 82.50 | 100.0 | 98.00 | 84.00 |
| | $GProd^V$ | 100.0 | 100.0 | 100.0 | **97.00** | 97.64 | **97.31** | **90.00** | 100.0 | 98.00 | 84.00 |

# 6 Experiments

To evaluate the empirical power of topological descriptors for graph products, we run three sets of experiments. The first investigates their effectiveness on BREC datasets [49] and minimal Cayley graphs [11] with varying number of nodes, which are popular benchmarks for assessing the expressive power of graph models. The second examines the runtime performance of the algorithms described in Theorems 4 and 5, using the BREC datasets. Finally, the third demonstrates how these descriptors can be integrated with GNNs for graph classification tasks.

**Expressivity.** For two input graphs $G$ and $H$, we use graph products to decide isomorphism as follows.

1. Compute the persistence diagrams for $G\square G$ and $G\square H$. If the diagrams differ, we conclude that $G$ and $H$ are *not isomorphic*. Otherwise, proceed to Step 2.
2. Compute the persistence diagrams for $G\square G$ and $H\square H$. If these diagrams differ, we again conclude that $G$ and $H$ are *not isomorphic*. If they are identical, the test is deemed inconclusive — i.e., $G$ and $H$ are considered isomorphic by this test.

Following this strategy, we compare our method (GProd) against standard PH (with vertex and edge filtrations), considering three filtration functions: edge closeness centrality, Jaccard index, and square clustering coefficients. Details regarding these functions are given in Appendix E.

Table 2 presents accuracy results. Overall, computing topological descriptors on graph products outperforms the standard approach (i.e., applying PH to the original graphs), although the two methods perform identically on several datasets. Even without introducing virtual nodes (see Corollary 1), the standard approach does not achieve higher accuracy in any of these experiments.

**Runtime.** For the vertex-level product filtration, we consider three algorithms: (i) our proposed method (Theorem 4); (ii) a naive computation on the full graph product using the union-find PH implementation described in Algorithm 1 of [26]; and (iii) a naive computation using the persistent homology routines from the gudhi library [39]. Similarly, for the edge-level product filtration, we compare three counterparts: our method (Theorem 5), a union-find PH implementation on the graph product, and the gudhi-based approach. We provide implementation details in Appendix E.

We evaluate all algorithms using BREC datasets. For each pair of graphs $G$ and $H$, we apply vertex/edge-level filtrations and compute the persistence diagrams of $G\square H$. Figure 3 reports the average runtime (in seconds, taken over 10 trials) each algorithm took to go over the entire dataset. Notably, as the graph sizes increase, Theorems 4 and 5 significantly outperform the other two methods.

**Graph classification.** We employ the following procedure for integrating descriptors based on graph products into GNNs for graph-level classification tasks. First, we select a diverse set of reference (or anchor) graphs, such as cycles, random graphs, and complete graphs. Then, for a given input graph $G$ and choice of filtration function, we either compute the product filtration between $G$ and each reference graph or obtain a filtration directly from their product. This process yields as many persistence diagrams as reference graphs. The resulting diagrams are then vectorized using an

---

[1]For full transparency, we note that Table 2 differs from earlier versions of the paper due to a bug in our code that incorrectly reported perfect accuracy for GProd. As a result, we revisited the experimental setup and considered alternative filtration functions and additional datasets to better illustrate the expressivity gains of our approach. We provide further details in the official code repository and in Appendix E.

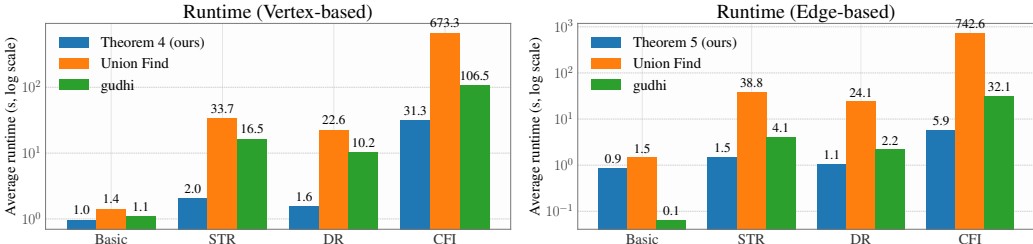

Figure 3: Average runtime (in sec.) of different implementations of topological descriptors for graph products. Our algorithms (Theorems 4 and 5) consistently outperform previous implementations.

appropriate embedding scheme [8]. Finally, the topological embeddings are concatenated with the GNN ones and passed through a classification head. In this work, we employ seven reference graphs generated using the NetworkX library [20]. Also, we apply the Gaussian vectorization scheme in [8]. We refer to Appendix E for further details.

In order to assess the effect of integrating topological descriptors, based on graph products, on the performance of GNNs, we consider two variants. `GProd` consists of computing PH on the box product of two graphs; `ProdFilt` consists of computing (vertex) product filtrations (as in Algorithm 1). In addition, we consider three GNN architectures: GIN [50], GCN [32], and GraphSAGE [21]. Each model uses a hidden dimension of 64, and the optimal number of layers (chosen from 2, 3, 5) is selected based on validation performance. For the graph product descriptors, we additionally treat the filtration functions (degree and betweenness centrality) as hyperparameters.

We consider six real-world popular datasets for molecular property prediction: COX2, DHFR, NCI1, NCI109, ZINC (12K), and PROTEINS [12, 30]. All tasks are binary classification, except ZINC, which is a regression dataset. For ZINC, we use the publicly available train/validation/test splits; for the remaining datasets, we adopt a random 80/10/10% split. Further details on datasets, hyperparameter selection, and model configuration are provided in the supplementary material. We compute the mean and standard deviation of the performance metrics (MAE $\downarrow$ for ZINC, and accuracy $\uparrow$ for all other datasets) over five runs with different seeds.

Table 3: Graph classification results. The inclusion of topological descriptors derived from graph products leads to consistent improvements in the predictive performance of GNNs.

| Model | DHFR $\uparrow$ | COX2 $\uparrow$ | PROTEINS $\uparrow$ | NCI1 $\uparrow$ | NCI109 $\uparrow$ | ZINC $\downarrow$ |
|---|---|---|---|---|---|---|
| GIN | $80.53 \pm 4.10$ | $76.60 \pm 2.61$ | $70.18 \pm 5.11$ | $80.19 \pm 1.83$ | $78.74 \pm 1.65$ | $0.57 \pm 0.02$ |
| GIN + GProd | $\mathbf{82.89} \pm 4.16$ | $78.30 \pm 5.90$ | $\mathbf{73.04} \pm 4.91$ | $\mathbf{81.65} \pm 2.42$ | $\mathbf{80.97} \pm 1.82$ | $\mathbf{0.54} \pm 0.02$ |
| GIN + ProdFilt | $81.58 \pm 2.79$ | $\mathbf{78.72} \pm 3.36$ | $72.50 \pm 3.54$ | $80.19 \pm 1.08$ | $80.15 \pm 2.47$ | $\mathbf{0.54} \pm 0.02$ |
| GCN | $78.68 \pm 5.54$ | $79.15 \pm 3.81$ | $73.04 \pm 2.92$ | $79.12 \pm 1.07$ | $79.08 \pm 1.95$ | $0.64 \pm 0.08$ |
| GCN + GProd | $\mathbf{83.95} \pm 5.38$ | $75.32 \pm 3.87$ | $73.75 \pm 5.70$ | $79.71 \pm 1.64$ | $80.00 \pm 1.10$ | $\mathbf{0.62} \pm 0.01$ |
| GCN + ProdFilt | $80.00 \pm 8.99$ | $\mathbf{80.85} \pm 6.73$ | $\mathbf{74.11} \pm 3.28$ | $\mathbf{79.90} \pm 0.82$ | $\mathbf{80.53} \pm 1.23$ | $\mathbf{0.62} \pm 0.01$ |
| SAGE | $\mathbf{84.21} \pm 3.48$ | $76.17 \pm 0.95$ | $72.86 \pm 1.62$ | $78.98 \pm 0.94$ | $\mathbf{79.81} \pm 1.31$ | $0.51 \pm 0.01$ |
| SAGE + GProd | $83.16 \pm 4.10$ | $\mathbf{80.85} \pm 2.61$ | $71.43 \pm 3.40$ | $\mathbf{80.29} \pm 0.75$ | $78.50 \pm 3.16$ | $\mathbf{0.48} \pm 0.01$ |
| SAGE + ProdFilt | $\mathbf{84.21} \pm 2.63$ | $77.87 \pm 4.41$ | $\mathbf{73.21} \pm 5.79$ | $80.00 \pm 1.09$ | $79.66 \pm 1.41$ | $\mathbf{0.48} \pm 0.01$ |

Table 3 reports the results. Overall, incorporating topological descriptors leads to best performance for most GNN/dataset combinations. Also, there is no clear winner between `GProd` and `ProdFilt`, and they yield the same performance on ZINC (the largest dataset). These experiments show that combining expressive topological features with GNNs can lead to consistent empirical improvements.

# 7 Conclusion and Broader Impact

We characterized the expressive power of EC on general color-based filtrations, and introduced a more efficient variant that has the same expressivity. We also examined the interaction of both EC and PH with graph box products, establishing contrasting results in terms of their respective benefits over the EC and PH of individual components. Furthermore, we devised efficient algorithms to compute persistence diagrams for a sub-collection of filtrations arising from the product of vertex-level, and edge-level, filtrations. Our experiments demonstrated consistent empirical gains. As graphs are ubiquitous in many real-world scenarios such as molecules, recommendation systems, and multi-way data, we envision that this work would open several avenues for applications in multiple areas.

## Acknowledgments and Disclosure of Funding

This research was conducted while the first author was participating during the 2024 Aalto Science Institute international summer research programme at Aalto University. We also acknowledge the computational resources provided by the Aalto Science-IT Project from Computer Science IT. We are grateful to the anonymous program chair, area chair and the reviewers for their constructive feedback and service. VG acknowledges the Academy of Finland (grant 342077), Saab-WASP (grant 411025), and the Jane and Aatos Erkko Foundation (grant 7001703) for their support. AS acknowledges the Conselho Nacional de Desenvolvimento Científico e Tecnológico (CNPq) (312068/2025-5). MJ would like to thank Sabína Gulčíková for her amazing help at Finland in Summer 2024. MJ would also like to thank Cheng Chen, Yifan Guo, Aidan Hennessey, Yaojie Hu, Yongxi Lin, Yuhan Liu, Semir Mujevic, Jiayuan Sheng, Chenglu Wang, Anna Wei, Jinghui Yang, Jingxin Zhang, an anonymous friend, and more for their incredible support in the second half of 2024.

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

# A Proofs

## A.1 Proofs for Section 2

We treat the proof of Proposition 1 separately in Appendix B.

## A.2 Proofs for Section 3

In this section, we will characterize the expressivity of EC on graphs. For the discussion of EC on simplicial complexes, please see Appendix D.

Before we state the proofs, we first remind the reader that in Section 3, we assumed that $G$ and $H$ have the same number of vertices (otherwise, they clearly are not isomorphic to each other).

**Remark 1.** *This assumption on $G$ and $H$ also gives a justification for why our definition of $\mathrm{EC}(G, f)^E$ does not include the Euler characteristic of $G_0^{f_e}$, as that is just the number of vertices on the graph.*

*Proof of Proposition 2.* We will first show that the second and third items are equivalent. For each vertex color function $f : X \to \mathbb{R}$, we use $g_f : X \times X \to \mathbb{R}$ to denote its correspondent edge color function.

Suppose $\mathrm{EC}^m(G, f)^E = \mathrm{EC}^m(H, f)^E$ for all possible $f : X \to \mathbb{R}$. Recall by assumption $G$ and $H$ have the same number of vertices $n$. Let $f$ be an vertex color function such that $0 < f(a) < f(b) < \min_{c_i \in C - \{a,b\}} f(c_i)$, then we have that

$$n - \#E_G(a, a) = \chi(G_a^{g_f}) = \chi(H_a^{g_f}) = n - \#E_H(a, a)$$

Hence $\#E_G(a, a) = \#E_H(a, a)$. By choosing another appropriate vertex color function, we can similarly show that $\#E_G(b, b) = \#E_H(b, b)$. Now, if we look back on the function $f$, we have that

$$n - \#E_G(a, a) - \#E_G(a, b) - \#E_G(b, b) = \chi(G_b^{g_f})$$

$$= \chi(H_b^{g_f}) = n - \#E_H(a, a) - \#E_H(a, b) - \#E_H(b, b).$$

This implies that $\#E_G(a, b) = \#E_H(a, b)$ since we already know that $\#E_G(a, a) = \#E_H(a, a)$ and $\#E_G(b, b) = \#E_H(b, b)$.

Conversely, suppose $\#E_G(a, b) = \#E_H(a, b)$ for all colors $a, b \in X$. Let $f$ be an edge color filtration function such that $0 < f(d_1) \le ... \le f(d_s)$, where $d_i$ is a relabeling of the colors $c_1, ..., c_s$ by this ordering.

Since $G$ and $H$ have the same number of vertices $n$, we have that $\chi(G_0^{g_f}) = n = \chi(H_0^{g_f})$. Thus, it suffices for us to show that $\chi(G_t^{g_f}) = \chi(H_t^{g_f})$ for all $t > 0$. Indeed,

$$\chi(G_t^{g_f}) = n - \sum_{i \le j \text{ s.t. } f(d_i), f(d_j) \le t} \#E_G(d_i, d_j)$$

$$= n - \sum_{i \le j \text{ s.t. } f(d_i), f(d_j) \le t} \#E_H(d_i, d_j)$$

$$= \chi(H_t^{g_f}).$$

Hence, we have that the second and third items are equivalent. Finally, we observe that clearly (1) implies (2) because the edge function induced by the max EC diagram is a special case of a symmetric edge coloring function. An argument very similar to the proof of (3) implies (2) will also show that (3) implies (1). This is because for all $t > 0$,

$$\chi(G_t^g) = n - \sum_{i \le j \text{ s.t. } g(d_i, d_j) \le t} \#E_G(d_i, d_j)$$

$$= n - \sum_{i \le j \text{ s.t. } g(d_i, d_j) \le t} \#E_H(d_i, d_j)$$

$$= \chi(H_t^g).$$

$\square$

*Proof of Proposition 3.* There is no difference between the definition of the 0-th dimensional component of the EC diagram and the max EC diagram, so the first two items are equivalent.

For (1) implies (3), we can choose any injective vertex color function such that $f$ obtains its minimum at the color $c$, Then $G^f_{f(c)}$ and $H^f_{f(c)}$ will be $G(c)$ and $H(c)$, and hence their Euler characteristics will be the same. For any $a \neq b \in X$, we can choose an injective vertex color function $f$ such that $f$ is smallest at $a$ and second smallest at $b$. In this case, we have that

$$\chi(G^f_{f(b)}) = \chi(H^f_{f(b)}).$$

Here we have that

$$\chi(G^f_{f(b)}) = \chi(G(a)) - \#E_G(a,b) + \chi(G(b)),$$

$$\chi(H^f_{f(b)}) = \chi(H(a)) - \#E_H(a,b) + \chi(H(b)).$$

Hence, we conclude that $\#E_G(a,b) = \#E_H(a,b)$ for all $a \neq b \in X$. The proof of (3) implies (1) is similar to that in Proposition 2. $\qquad\square$

*Proof of Theorem 2.* The first two statements come directly from combining the statements of Proposition 2 and Proposition 3. The third statement follows from the observation that $\chi(G(c)) + \#E_G(c,c) = \#V_G(c)$. $\qquad\square$

## A.3 Proofs for Section 4

*Proof of Proposition 4.* The proof for why edge-level PH can differentate $G\square G$ and $H\square H$ is mainly self-explanatory by the caption of Figure 1. The reason why edge-level PH is able to differ $G\square G$ and $H\square H$ is because when adding the edges between $(\text{blue}, \text{blue})$ and $(\text{red}, \text{blue})$ creates a cycle for $G\square G$ and does not change the 1-dimensional persistence diagram for $H\square H$.

To see why PH cannot tell apart $G$ and $H$ individually. We first note that there are no cycles, so we only need to focus on the 0-th dimensional components.

**For vertex-level PH:** Write $r = f_v(\text{red})$ and $b = f_v(\text{blue})$. If $r = b$, then their persistence diagrams are the same since $G$ and $H$ have the same number of vertices and connected components. If $b > r$, then the subgraphs $G_r$ and $H_r$ are exactly the same graph, so PH cannot tell that step apart. At $t = b$, for both $G$ and $H$, two blue vertices are born and deceased at the same time, and one red vertex is also killed. If $r > b$, then again $G_b$ and $H_b$ are isomorphic graphs. At $t = r$, for both $G$ and $H$, two red vertices are born and dead at the same time, and one blue vertex is also killed. Thus, we conclude the vertex-level PH cannot tell apart $G$ and $H$.

**For edge-level PH:** Write $a = f_e(\text{red}, \text{red})$ and $b = f_e(\text{red}, \text{blue})$. If $a = b$, then they have the same persistence diagrams because $G$ and $H$ have the same number of vertices and connected components. If $a > b$, then there are two deaths that occur at $t = b$ and one death at $t = a$ for both graphs. If $b > a$, then there is one at $t = a$ and then two deaths are $t = b$ for both graphs. Thus, we conclude the edge-level PH cannot tell apart $G$ and $H$. $\qquad\square$

*Proof of Proposition 5.* For two general colored graphs $A$ and $B$. For colors $a, b \in X$, we observe that $\#V_{A\square B}((a,b)) = \#V_A(a)\#V_B(b) + \#V_B(a)\#V_A(a)$ (here the second term is due to the symmetry assumption).

For colors $(a,b), (c,d) \in X \times X$, we also wish to enumerate $\#E_{A\square B}((a,b),(c,d))$. Indeed, without loss when both $a \neq c$ and $a \neq d$, the definition of box product indicates that $\#E_{A\square B}((a,b),(c,d)) = 0$. The axioms for the box product indicates that edges can only occur between two pairs of colors that share at least one color in common. By rearranging the order of tuples if necessary, we without loss only need to enumerate $\#E_{A\square B}((a,b),(a,d))$. If $b \neq d$, then the number is given exactly as

$$\#E_{A\square B}((a,b),(a,d)) = \#V_A(a)\#E_B(b,d) + \#V_B(a)\#E_A(b,d).$$

Now since $G$ and $H$ in the statement of the proposition have the same EC diagram for all possible filtrations, Theorem 2 implies that for all $c_1, c_2 \in X$, $\#V_G(c_1) = \#V_H(c_1)$ and $\#E_G(c_1,c_2) = \#E_H(c_1,c_2)$. Plugging this equality into the formulas derived above, we have that

$$\#V_{G\square G}((a,b)) = \#V_{G\square H}((a,b)) = \#V_{H\square H}((a,b)),$$

$$\#E_{G \square G}((a,b),(c,d)) = \#E_{G \square H}((a,b),(c,d)) = \#E_{H \square H}((a,b),(c,d))$$

for all possible choices of colors. Thus, Theorem 2 implies that $G \square G, G \square H$, and $H \square H$ all have the same EC diagrams for all possible filtrations. □

*Proof of Corollary 1.* We first observe that for any two graphs $A$ and $B$, we have that

$$A_+ \square B_+ = (A \sqcup B \sqcup A \square B)_+.$$

When $A = B = G$ from the statement of this corollary, we observe that

$$G_+ \square G_+ = (G \sqcup G \sqcup G \square G)_+.$$

Write the two disjoint copies of $G$ produced here as $G_0$. Observe that each vertex $v \in G_0$ (or $G_1$) has the color $(c(v), \text{virtual})$. Observe that no vertices in $G \square G$ has the color that has the tag `virtual` in it, so we can choose a coloring procedure such that we go through the filtrations on $G_0$ and $G_1$ first before going through $G \square G$. Running vertex-level (resp. edge-level PH) through this coloring procedure, we will produce two copies of vertex-level PH (resp. edge-level PH) for $G$, which recovers the original information. A similar procedure also works for $H$. Thus, the PH of $G_+ \square G_+, G_+ \square H_+, H_+ \square H_t$ has at least as much information as the PH of $G$ and $H$. The part for "strictly more information" follows from Proposition 4. □

## A.4 Proofs for Section 5

*Proof of Proposition 6.* To check that the graphs are the same, we wish to check that they have the same vertex set and edge sets. Indeed, let $t \in \mathbb{R}$, then

$$V(G_t^{f_G} \square H_t^{f_H}) = \{(g,h) \mid f_G(c(g)) \le t \text{ and } f_H(c(h)) \le t\}$$
$$= \{(g,h) \mid \max(f_G(c(g)), f_H(c(h))) \le t\} = \{(g,h) \mid F((g,h)) \le t\} = V((G \square H)_t).$$

Now to check edges, we have that for $(g_1, h_1), (g_2, h_2) \in V(G_t \square H_t) = V((G \square H)_t)$. Now suppose there is an edge between $(g_1, h_1)$ and $(g_2, h_2)$ in $(G \square H)_t^f \subseteq G \square H$. Since this is an edge in $G \square H$ it means either of two following cases

1. $g_1 = g_2$ and $h_1 \sim h_2$ in $H$. Now if we can show that $h_1 \sim h_2$ in $H_t$, then this will give an edge between $(g_1, h_1)$ and $(g_2, h_2)$ in $G_t^{f_G} \square H_t^{f_H}$. Indeed, that just comes from the definition of a vertex filtration as the colors on $h_1, h_2$ are both less than or equal to $t$ under $f_H$.

2. $h_1 = h_2$ and $g_1 \sim g_2$ in $G$. The argument is nearly identical to the first case.

Conversely, suppose there is an edge between $(g_1, h_1)$ and $(g_2, h_2)$ in $G_t^{f_G} \square H_t^{f_H}$. Then, since $F((g_1, h_1)) \le t$ and $F((g_2, h_2)) \le t$ (as they have the same vertex set), it follows that this edge has to be in $(G \square H)_t$ by definition of vertex filtration. □

*Proof of Proposition 7.* To check that the graphs are the same, we wish to check that they have the same vertex set and edge sets for all $t \ge 0$. The vertex set at time $t$ for $t \ge 0$ is always the collection of all vertices. The proof for the edge sets follows similarly to the proof in Proposition 6. □

Now we will prove that the outputs of Algorithm 1 and Algorithm 2 gives the 0-dimensional PH diagrams for the product of vertex-level filtraitons and edge-level filtrations respectively. For Theorem 4 and Algorithm 1, we also note that there is a more symmetric presentation of Algorithm 1 in terms of Algorithm 3, which we will show gives the same output. We now proceed to prove Theorem 4.

*Proof.* **For Algorithm Output:** We first observe that if we modify nt_deaths in Algorithm 1 to be defined as nt_deaths $= \text{betti}_G[i + 1] \times \text{deaths}_H[i + 1]$, then it would give the same output. This is because every vertex in $\text{births}_G[i + 1]$ is in $\text{betti}_G[i + 1]$ (with compatible representatives), and subtracting the two versions of nt_deaths from nt_births would still give the same set (since the intersection of nt_births with their respective complements are the same).

From here, showing that Algorithm 1 produces the correct output is equivalent to showing the following theorem.

**Theorem 6.** *In the set-up of Proposition 6, mark all the times $G_t \square H_t$ changes as $a_0 < ... < a_n$. Let $b_i(G)$ and $d_i(G)$ be the number of non-trivial births at $a_t$ and the number of non-trivial deaths at $a_t$ (and similarly for $H$). The 0-dimensional PH of the filtration may be computed inductively by the following procedure.*

*For time after $-\infty$ to $a_0$, initialize the list with $\beta_0(G_{a_0})\beta_0(H_{a_0})$ tuples of the form $(a_0, -)$. Each tuple represents a connected component $X_k \square Y_j$ where $X_k$'s and $Y_j$'s are the connected components of $G_{a_0}$ and $H_{a_0}$ respectively. Then mark the remaining tuples as trivial holes, ie. $(a_0, a_0)$.*

*For time after $a_i$ to $a_{i+1}$, $i \geq 0$:*

1. *If $H_{a_{i+1}} = H_{a_i}$, then there are exactly $\beta_0(H_t)b_{i+1}(G)$ non-trivial births of tuples of the form $(a_{i+1}, -)$. The non-trivial deaths that die at $a_{i+1}$ are exactly as follows - for each component $Y_j$ of $H_{a_i}$, and for each component $X_k$ of $G_{a_i}$, the component $(X_k, Y_j)$ dies for all $X_k$'s that die in the filtration $G_{a_i} \subset G_{a_{i+1}}$.*
2. *If $G_{a_{i+1}} = G_{a_i}$ then a similar statement holds by symmetry.*
3. *In the general case, we refine the filtration of $G \square H$ from $a_i$ to $a_{i+1}$ into a two-step filtration:*

$$G_{a_i} \square H_{a_i} \overset{(A)}{\subset} G_{a_{i+1}} \square H_{a_i} \overset{(B)}{\subset} G_{a_{i+1}} \square H_{a_{i+1}},$$

*and apply Step 1 to (A) and Step 2 to (B).*

 ***Note*** *that when applying Step 2 to (B), "some of the non-trivial deaths produced are trivial deaths". The reason why is, although in **algorithmic times**, there may be vertices that are born in $G_{a_{i+1}} \square H_{a_i}$ and die in $G_{a_{i+1}} \square H_{a_{i+1}}$, Steps (A) and (B) both happened at the same time during **filtration time**.*

 *Thus, we need to retroactively go back and mark all vertices born in $G_{a_{i+1}} \square H_{a_i}$ and die in $G_{a_{i+1}} \square H_{a_{i+1}}$ as trivial pairs. In total, this incurs a subtraction of $b_{i+1}(G)d_{i+1}(H)$ (where $b_{i+1}(G)$ is the number of non-trivial births from $G_{a_i} \subset G_{a_{i+1}}$ and $d_{i+1}(H)$ is the number of non-trivial deaths from $H_{a_i} \subset H_{a_{i+1}}$), with clear vertex representatives, from the components produced to be counted as trivial holds.*

4. *The remaining tuples produced but not in the discussion above are all trivial holes.*

*Proof of Theorem 6.* For two arbitrary simple graphs $A$ and $B$, we first observe note that $\beta_0(A \square B) = \beta_0(A)\beta_0(B)$. This can either be seen directly from the definition or by noting that $A \square B$ is exactly the 1-skeleton of the topological product $A \times B$ (equipped with the canonical cell structure). Since removing 2-cells does not change connectivity, it follows that $H_0(A \square B) \cong H_0(A \times B)$, and the rank of the latter can be computed by the Kunneth formula to be $\beta_0(A)\beta_0(B)$. (Note that we are able to use the Kunneth formula here because the integral homologies of a (finite) graph are all finitely generated free abelian groups, so the rank does not change when switching to a field).

Let us prove the theorem by induction. Our inductive hypothesis will be that the $k$-th step of the algorithm agrees with the PH diagram at the stage $G_{a_k} \square H_{a_k}$. Also note that Step 3 subsumes Step 1 and 2 as special cases, so in our proof we only need to check Step 3 and 4. By convention, we write $a_{-1} = -\infty$. When $k = -1$, clearly this holds since the lists are both empty. At $k = 0$, the discussion in the paragraph above shows that it agrees with the PH diagram calculation.

Now by induction suppose we have that the result matches with PH up to $k = i \geq 0$, then we wish to show this is also the case for $k = i + 1$. By the algorithm, this step of the filtration is divided into

$$G_{a_i} \square H_{a_i} \overset{(A)}{\subset} G_{a_{i+1}} \square H_{a_i} \overset{(B)}{\subset} G_{a_{i+1}} \square H_{a_{i+1}}.$$

Observe that this refinement clearly does not change the PH diagram (since the PH diagram is independent of the over of attaching new vertices and edges in one step of the color-based filtration). Thus, we only need to show that the algorithm the Theorem provides behaves the same as PH in both step (A) and (B).

Let us first examine (A). Now observe that $\beta_0(G_{a_{i+1}} \square H_{a_i}) - \beta_0(G_{a_i} \square H_{a_i}) = (\beta_0(G_{a_{i+1}}) - \beta_0(G_{a_i}))\beta_0(H_{a_i}) = (b_{i+1}(G) - d_{i+1}(G))\beta_0(H_{a_i})$. On the other hand, we also have that $\beta_0(G_{a_{i+1}} \square H_{a_i}) - \beta_0(G_{a_i} \square H_{a_i}) = b_{i+1}(G \square H) - d_{i+1}(G \square H)$. Thus, if we could show that $d_{i+1}(G \square H) = d_{i+1}(G)\beta_0(H_{a_i})$, it also follows that $b_{i+1}(G \square H) = b_{i+1}(G)\beta_0(H_{a_i})$ (which proves the birth times). Since there are already $b_{i+1}(G)\beta_0(H_{a_i})$ components created in the filtration

step (A) by the product, this would show that this constitutes all the non-trivial births (in particular this allows an identification of the component with the birth time).

Thus it suffices for us to show that the deaths occur exactly in the following form - for each component $Y_j$ of $H_{a_i}$, and for each component $X_k$ of $G_{a_i}$, the component $(X_k, Y_j)$ dies for all $X_k$'s that die in the filtration $G_{a_i} \subset G_{a_{i+1}}$. Now because the component from $H$ does not change in this step, it is clear that there are at least $d_{i+1}(G)\beta_0(H_{a_i})$ such many deaths coming from the description above. What we want to check is that these are the only deaths that occur. Since we accounted for all possible merging of $(X_{k_1}, Y_j)$ and $(X_{k_2}, Y_j)$, another death that could occur has to be of the form to merge $(X_k, Y_{j_1})$ and $(X_k, Y_{j_2})$ (by the virtue of box product). However, for such death to occur, it must mean that $H$ has changed in the filtration, which is a contradiction. Thus, we have identified all the deaths in the filtration step (A).

The case for (B) follows by symmetry, and we gave an extensive discussion in the theorem statement on why some non-trivial deaths would be marked as trivial deaths. Finally, by induction, we have proven the theorem. □

**Equivalence between Algorithm 1 and Algorithm 3:** It suffices to show that the list nt_births produced for both Algorithm 1 and Algorithm 3 are the same. Indeed, at index $i$, nt_births produced by Algorithm 3 is exactly the set $\{(v, w) \in G \square H \mid ((1) \text{ or } (2)) \text{ and } ((3) \text{ or } (4))\}$, where:

- (1) $v$ is born non-trivially at $a_i$ and $w$ is still alive at $a_{i-1}$.

- (2) $v$ is still alive at $a_i$ and $w$ is born non-trivially at $a_i$.

- (3) $v$ is still alive at $a_{i-1}$ and $w$ is born non-trivially at $a_i$.

- (4) $v$ is born non-trivially at $a_i$ and $w$ is still alive at $a_i$.

On the other hand, the list nt_births produces for Algorithm 1 is exactly the set $\{(v, w) \in G \square H \mid (\dagger)\}$, where:

- ($\dagger$) is the condition that - (1) or (2) holds, but excluding the pairs $(v, w)$ where $v$ is born non-trivially at $a_i$ and $w$ dies at $a_i$.

Now suppose $(v, w)$ is in nt_births for Algorithm 3, so (1) or (2) holds. Suppose for contradiction $v$ is born non-trivially at $a_i$ and $w$ dies at $a_i$, then the pair $(v, w)$ is false for the statement (3) or (4), since both (3) and (4) require $w$ to still be alive at $a_i$. Thus, $(v, w)$ is in nt_births for Algorithm 1.

Now suppose $(v, w)$ is in nt_births for Algorithm 1, so (1) or (2) holds. We wish to show that $(v, w)$ also satisfies (3) or (4). Indeed, since $(v, w)$ is not excluded, this means that either (5) $v$ is not born non-trivially at $a_i$ or (6) $w$ does not die at $a_i$. In this case, we see that:

- If (1) holds, then (5) is not possible, so (6) has to hold, which implies (4) holds.

- If (2) holds, then (6) always holds, and $w$ is born non-trivially at $a_i$. If $v$ is born before $a_i$, then (3) holds. If $v$ is born at $a_i$, it is a non-trivial birth by (2) and (4) holds since $w$ is still alive at $a_i$.

**For Runtime:** We can compute the two individual persistence diagrams in $O(m_G \log(m_G) + n_G)$ and $O(m_H \log(m_H) + n_H)$ time. We can pre-compute the relevant data from the diagrams. If we assume the coloring set has constant size, then filtration steps are also constant, so this can be done in the runtime dominated by the result of the theorem. We initialize the persistence diagram with $n_G n_H$ tuples with empty entries. In this case, when we loop through the filtration steps, our (possibly unoptimized) algorithm implements the intermediate steps in the Theorem by looping through the $n_G n_H$ tuples. This yields the runtime as $O(n_G n_H + m_G \log m_G + m_H \log m_H + n_G + n_H)$. □

Now we will look of the algorithm for the product of two edge-level filtrations. Let us first rewrite the algorithmic description in Algorithm 2 into a more mathematical formulation below.

**Algorithm 3** Computing the 0-dim PH Diagrams in Product of Vertex-Level Filtrations (Symmetric)

**Input:** The vertex-level filtrations $(G, f_G)$ and $(H, f_H)$
**Output:** The 0-dim PH diagram for $(G \square H, f_G \square f_H)$

1: $\text{impl}_G, \text{impl}_H \leftarrow$ 0-dim PH diagrams for $(G, f_G)$ and $(H, f_H)$.
2: filtration_steps $\leftarrow \{-\infty\} \cup$ filtration steps for $f_G$ and $f_H \cup \{+\infty\}$, and ordered.
3: $\text{births}_G, \text{births}_H \leftarrow$ list of tuples born non-trivially in $G$ (resp. $H$) at each time in filtration_steps.
4: $\text{betti}_G, \text{betti}_H \leftarrow$ list of vertices alive representing components of the subgraph of $G$ (resp. $H$) at each time in filtration_steps.
5: $\text{deaths}_G, \text{deaths}_H \leftarrow$ list of tuples that die non-trivially in $G$ (resp. $H$) at each time in filtration_steps.
6: Remove $\{-\infty\}$ from filtration_steps.
7: $\textbf{output} \leftarrow [(i, j, \max(f_G(i), f_H(i)), \text{None}) \textbf{ for } (i, j) \text{ in } V(G) \times V(H)]$.
8: **for** i in [0, len(filtration_steps)] **do**
9:     $a_i \leftarrow$ filtration_steps[i].
10:     **for** v in $\text{deaths}_H[i + 1]$ **do**                  ▷ Marking Non-Trivial Deaths
11:         Mark all tuples $p$ in **output** with $p[1] = v, p[2] < a_i, p[3] = \text{None}$ with $p[3] \leftarrow a_i$.
12:     **for** w in $\text{deaths}_G[i + 1]$ **do**                 ▷ Marking Non-Trivial Deaths
13:         Mark all tuples $p$ in **output** with $p[0] = w, p[2] < a_i, p[3] = \text{None}$ with $p[3] \leftarrow a_i$.
14:     nt_births1 $\leftarrow \text{births}_G[i + 1] \times \text{betti}_H[i] + \text{betti}_G[i + 1] \times \text{births}_H[i + 1]$.
15:     nt_births2 $\leftarrow \text{betti}_G[i] \times \text{births}_H[i + 1] + \text{births}_G[i + 1] \times \text{betti}_H[i + 1]$.
16:     nt_births $\leftarrow$ nt_births1 $\cap$ nt_births2.
17:     Mark all tuples $p$ in **output** with $(p[0], p[1]) \notin$ nt_births, $p[2] = a_i, p[3] = \text{None}$ with $p[3] \leftarrow a_i$.               ▷ Marking Trivial Deaths
18: **return** [(p[2], p[3]) **for** p in **output**]

---

**Theorem 7.** *In the set-up of Proposition 7, mark all the time $G_t \square H_t$ changes as $a_0 = 0 < a_1 < ... < a_n$. The 0-dimensional persistence diagram of $(G \square H)_t = G_t \square H_t$ with respect to $f$ can be computed by the following procedure:*

*1. All vertices are born at time $0$.*
*2. For $i \geq 0$, for each filtration step $G_{a_i} \square H_{a_i} \subseteq G_{a_{i+1}} \square H_{a_{i+1}}$, we refine the step as*

$$G_{a_i} \square H_{a_i} \overset{(A)}{\subset} G_{a_{i+1}} \square H_{a_i} \overset{(B)}{\subset} G_{a_{i+1}} \square H_{a_{i+1}}.$$

*3. In Step (A), there are $h_i^b \cdot g_{i+1}^d$ deaths. Here $h_i^b$ is the number of vertices alive in $H_{a_i}$ and $g_{i+1}^d$ are the number of vertices that die in the step $G_{a_i} \subset G_{a_{i+1}}$.*
*4. In Step (B), there are $g_{i+1}^b \cdot h_{i+1}^d$ deaths. Here $g_i^b$ is the number of vertices alive in $G_{a_{i+1}}$ and $h_{i+1}^d$ is the number of vertices that die in the step $H_{a_i} \subset H_{a_{i+1}}$.*
*5. The remaining vertices after $a_n$ all die at $\infty$.*

*The runtime is $O(\max(n_G \log n_G + n_H \log n_H, m_G \log m_G + n_G, m_H \log m_H + n_H))$.*

Evidently, we can see that Algorithm 2 outlined in Theorem 5 is equivalent to the description of Theorem 7. Thus, it suffices for us to prove Theorem 7. Before this, we first give a concrete example of using Theorem 7.

**Example 2.** *Consider the setup of $G = H$ given in Figure 4, and so $a_0 = 0, a_1 = 1, a_2 = 2$. Note that $G_0$ is the discrete graph with 2 red nodes and 1 blue node. $G_1$ connects an edge between 1 red node and 1 blue node. $G_2$ connects the last remaining edge.*

*We wish to compute the PH of the product of edge filtrations on $G \square H$. At $t = 0$, all vertices are spawned, so there are 9 pairs of tuples of the form $(0, -)$. At $t = 1$, we see that 5 vertex deaths occur during this time. This number can be seen from the algorithm as*

$$h_0^b \cdot g_1^d + g_1^b \cdot h_1^d = (3) \cdot (1) + (2) \cdot (1) = 5.$$

*Thus, we mark 5 of the 9 tuples as $(0, 1)$. At $t = 2$, we see that 3 vertex deaths occur during this time. This can be computed from the algorithm as*

$$h_1^b \cdot g_2^d + g_2^b \cdot h_2^d = (2) \cdot (1) + (1) \cdot (1) = 3.$$

*Thus, we mark 3 of the remaining tuples as $(0,2)$. For $t > 2$, there is 1 vertex left, so we mark the last tuple as $(0,\infty)$.*

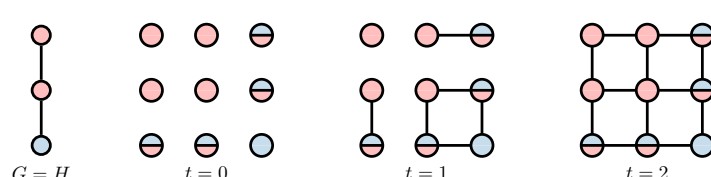

$$f_e(\mathrm{red}, \mathrm{red}) = 2 > f_e(\mathrm{red}, \mathrm{blue}) = 1$$

$G = H$      $t = 0$      $t = 1$      $t = 2$

Figure 4: The product of the edge filtrations on $G = H$ given by $f_e(\mathrm{red}, \mathrm{red}) = 2 > f_e(\mathrm{red}, \mathrm{blue}) = 1$. See Example 2 for how the algorithm in Theorem 5 is used to compute the PH of this filtration.

We now proceed with the proof.

*Proof of Theorem 7.* **For Algorithm Output:** Step (1) is obvious. By the well-definedness of PH, the number of deaths in the filitration $G_{a_i} \Box H_{a_i} \subset G_{a_{i+1}} \Box H_{a_{i+1}}$ is the sum of deaths in the two step refinement given. Thus, it suffices for us to verify Step (A) gives the correct number of deaths, since the case for Step (B) follows by a symmetric argument.

For the ease of notations, let us write $n = h_i^b$ and $m = g_{i+1}^d$. We seek to show that the number of deaths in the step $G_{a_i} \Box H_{a_i} \overset{(A)}{\subset} G_{a_{i+1}} \Box H_{a_i}$ is $nm$.

Indeed, since the number of vertices still alive in $H_{a_i}$ corresponds to its number of connected components, we can write $C_1, ..., C_n$ as the connected components of $H_{a_i}$. Now we have that,

$$G_{a_i} \Box H_{a_i} = G_{a_i} \Box \left( \bigsqcup_{j=1}^{n} C_j \right) = \bigsqcup_{j=1}^{n} G_{a_i} \Box C_j \text{ and } G_{a_{i+1}} \Box H_{a_i} = \bigsqcup_{j=1}^{n} G_{a_{i+1}} \Box C_j.$$

The number of vertices that dies in Step (A) is additive under this disjoint union, so it suffices for us to show that the number of vertices that die in the step $G_{a_i} \Box C_j \subset G_{a_{i+1}} \Box C_j$ is $m$. Now indeed, write $D_1, ..., D_\ell$ as the connected components of $G_{a_i}$. The connected components $D_j$ and $D_k$ are merged in the step $G_{a_i} \subset G_{a_{i+1}}$ if and only if the connected components $D_j \Box C_i$ and $D_k \Box C_i$ are merged. This proves the desired claim.

**For Runtime:** We see that Algorithm 2 does not need to compute the graph product and is dependent on the PH for $G$ and PH for $H$ separately. As we discussed in the paragraph under Theorem 4, the runtime to precompute the PH diagrams for $G$ and $H$ respectively are $O(m_G \log(m_G) + n_G)$ and $O(m_H \log(m_H) + n_H)$. We can precompute the number of non-trivial deaths and number of vertices still alive in $O(n_G \log(n_G))$ and $O(n_H \log(n_H))$ time by sorting the two (0-dim) PH diagrams by death-times and count accordingly. Then we loop through both lists in parallel to compute the diagram (which is in linear time). Thus, the runtime of Theorem 5 would be $O(\max(n_G \log(n_G) + n_H \log(n_H), m_G \log(m_G) + n_G, m_H \log(m_H) + n_H))$. $\qquad \Box$

Finally, we discuss the algorithm for the 1-dimensional PH diagrams of product of filtrations.

*Proof of Proposition 8.* Since a cycle born in a graph never dies, the 1-dimensional persistence diagrams are always of the form $(b(e), \infty)$ where $e$ represents the edge that creates an independent cycle. To track the evolution of the 1-dimensional persistence diagrams for the product of any filtration, it suffices for us to track how the first Betti number changes. The number of tuples created is the difference between successive $\beta_1$'s.

Thus, it suffices for us to verify the formula for $\beta_1(A \Box B)$. The box product $A \Box B$ is exactly the 1-skeleton of the topological product $G \times H$, endowed with the canonical CW complex structure

from both $A$ and $B$. The Euler characteristic of the product of two finite CW complexes is the product of their Euler characteristics, so we have that $\chi(A \times B) = \chi(A)\chi(B)$ (see Exercise 2.2.20 of [22]). On the other hand, since $A\square B$ is the 1-skeleton of $A \times B$, we have that

$$\chi(A\square B) = \chi(A \times B) - \#E(A)\#E(B).$$

On the other hand we have that

$$\chi(A\square B) = \beta_0(A\square B) - \beta_1(A\square B),$$

and hence we have that

$$\beta_1(A\square B) = \#E(A)\#E(B) + \beta_0(A\square B) - \chi(A)\chi(B).$$

Since removing 2-cells do not affect the connectivity of the complex, we have that $\beta_0(A\square B) = \beta_0(A \times B) = \beta_0(A)\beta_0(B)$, where the last equality follows from Kunneth's formula. This concludes the proof.

**For Runtime**: For $G$ and $H$ graphs, we see that the quantity $\beta_1(G\square H)$ can be expressed in terms of (1) the number of edges, (2) the number of connected components, and (3) the number of vertices of G and H respectively. To compute the 1-dim persistence diagram of a product filtration, it suffices for us to keep track of how (1), (2), (3) changes over time. These can all be done in a time dominated by $O(m_G \log m_G + n_G + m_H \log m_H + n_H)$ (ie. the time to compute the persistence diagrams on the two separate graphs). $\qquad\square$

## B   A Max Version of Persistent Homology

In this section, we discuss a max version of persistent homology where the edge-level filtration function $f_e$ in PH is determined by the vertex-level filtration function $f_v$. We will show that the edge-level PH of max PH can be computed directly from the vertex-level PH. More precisely, we reformulate the first part of Proposition 1 more generally to the following proposition:

**Proposition 9.** *Let $f_v$ be a vertex-level filtration on $G$. Consider an edge-level filtration $f_e$ on $G$ with*

*1. $F_e(w)$ is constant for all $w \in V(G)$ and is less than the minimum of $F_v$ on $V(G)$.*
*2. $F_e(w, w') := \max(f_v(w), f_v(w'))$ for all $(w, w') \in E(G)$.*

*The vertex death-times of $f_e$ are the same as the death-times of $f_v$. Furthermore, if the edge-level filtration is given a decision rule to kill off the vertex that has higher value under $f_v$ (and we arbitrarily choose which vertex to kill off in a tie), it may be chosen so that the same vertices die at the same time for both filtrations.*

*Proof.* To show that the death-times of $f_e$ are the same as that of $f_v$, it suffices to check that at each time step $t$, the number of vertices that die on both filtrations (notationally we write $D_v(t)$ and $D_e(t)$) are the same. Indeed, write the possible values of $f_v$ in order as $a_1 < ... < a_n$. Outside of this list of values, we clearly have that $D_v(t) = D_e(t) = 0$.

At $t = a_1$, the vertex level filtration creates $G(a_1)$ (see Section 3 to see the notation) and all the deaths can only happen in $G(a_1)$. At the same time, observe that all the edges spawned at time $t = a_1$ for edge level filtration happens only in the subgraph of $V_G(a_1)$ and gives the full subgraph $G(a_1)$. Thus, we have that $D_v(a_1) = D_e(a_1)$. Furthermore in the first step, we could clearly choose the same vertices to kill off for both filtrations.

Now suppose by induction we have that at $t = a_{i-1}$, the previous death times have been the same and that we have chosen the same vertices to kill off for both filtrations. Now at $t = a_i$, the vertex level filtration creates $\bigcup_{j=1}^{i} E_G(a_j, a_i)$, and the edge level filtration also adds all the edges in $\bigcup_{j=1}^{i} E_G(a_j, a_i)$. For our purposes, the vertex level filtration will first spawn all of $G(a_i)$ first (before adding the edges in $E_G(a_j, a_i)$ for $j < i$), and the edge level filtration will first add in all of the edges in $G(a_i)$ similarly. In this case, clearly we can still choose the same vertices to kill off for both filtrations by the inductive hypothesis. Doing this up to $a_n$ concludes the proof. $\qquad\square$

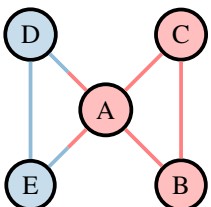

Figure 5: Graph $G$ in Example 3.

The second part of Proposition 1 can be more generally reformulated to the following statement.

**Proposition 10.** *Let $f_v$ be a vertex-level filtration on $G$. Consider the same edge-level filtration $f_e$ given in Proposition 9. The cycle birth times of the vertex-level filtration are the same as the cycle birth-times of the edge-level filtration.*

*Proof.* This is because the birth-time of the edges are the same in both the vertex-level filtration and the edge-level filtration. Since color-based PH is independent of the order in adding in the simplicies marked with the same color, we can add the edges in both filtrations in order such that a cycle occurs in the vertex-level filtration if and only if it occurs in the edge-level filtration. A similar proof as in the previous proposition shows that the creation of cycles would be the same. $\qquad\square$

## C An Example Computing EC

Here we give an explicit example computing the EC diagram.

**Example 3.** *Consider the graph $G$ in Figure 5 with $f_v(\text{red}) = 1$, $f_v(\text{blue}) = 2$, $f_e(\text{red}, \text{red}) = 1$, $f_e(\text{blue}, \text{blue}) = 2$, $f_e(\text{red}, \text{blue}) = 3$. We have that*

$$\text{EC}(G, f)^V = \chi(G_1^{f_v}), \chi(G_2^{f_v}) = 0, -1 \text{ and } \text{EC}(G, f)^E = \chi(G_1^{f_e}), \chi(G_2^{f_e}), \chi(G_3^{f_e}) = 2, 1, -1.$$

*Proof for Example 3.* Consider the set-up in the example again, recall this gives a graph $G$ with $f_v(\text{red}) = 1$, $f_v(\text{blue}) = 2$, $f_e(\text{red}, \text{red}) = 1$, $f_e(\text{blue}, \text{blue}) = 2$, $f_e(\text{red}, \text{blue}) = 3$. We wish to show that

$$\text{EC}(G, f)^V = \chi(G_1^{f_v}), \chi(G_2^{f_v}) = 0, -1 \text{ and } \text{EC}(G, f)^E = \chi(G_1^{f_e}), \chi(G_2^{f_e}), \chi(G_3^{f_e}) = 2, 1, -1.$$

Now to compute $\text{EC}(G, f)^V$, we observe that $G_1^{f_v}$ is the subgraph of $G$ generated by red vertices (A, B, C), so it is the cycle graph of 3-vertices. Thus, $\chi(G_1^{f_v}) = 0$. When $t$ reaches 2, we have that $G_2^{f_v} = G$, so the Euler characteristic is $-1$.

Now to compute $\text{EC}(G, f)^E$, we note that at $t = 0$, $G_0^{f_e}$ is the discrete graph on 5 vertices. At $t = 1$, all the red edges are added in, so $G_1^{f_e}$ is the disjoint union of 2 vertices and the cycle graph of 3-vertices. Thus, $\chi(G_1^{f_e}) = 0 + 2 = 2$. At $t = 2$, a single blue edge is added between the two disjoint vertices, so $\chi(G_2^{f_e}) = \chi(G_1^{f_e}) - 1 = 1$. At $t = 3$, two more edges are added and hence $\chi(G_3^{f_e}) = 1 - 2 = -1$. $\qquad\square$

## D Expressivity of EC on Simplicial Complexes

Let $K$ be a simplicial complex of dimension $n$ where each vertex is assigned a color in some coloring set $X$. The Euler characteristic of $K$ is defined by

$$\chi(K) = \sum_{i=0}^{n} (-1)^n \#\{\text{simplicies in } K \text{ of dimension } i\}.$$

**Notation:** Let $c_1, ..., c_i \in X$ be a list of distinct colors with $i \leq j$, we define $S_K^j(c_1, ..., c_i)$ as the number of $j$-simplices in $K$ whose vertice's set of colors is $\{c_1, ..., c_i\}$.

Under suitable extension of the definition of EC and max EC diagrams to a simplicial complex, we can, in fact, characterize them completely in terms of the combinatorial data provided by $S_K^{\cdot j}(c_1, ..., c_i)$. We will make these definitions precise and in particular prove the following theorem.

> **Theorem 8.** *Let $K, M$ be simplicial complexes of dimension $n$. The following are equivalent:*
> 1. $\text{EC}(K, f_0, ..., f_n) = \text{EC}(M, f_0, ..., f_n)$ *for all possible $f_0, ..., f_n$. Here, each $f_i$ is an $i$-simplex coloring function, to be elaborated upon in Appendix D.*
> 2. $\text{EC}^m(K, f_0) = \text{EC}^m(M, f_0)$ *for all possible $f_0$.*
> 3. *For all $j \leq n$, $S_K^j(c_1, ..., c_i) = S_M^j(c_1, ..., c_i)$ for all distinct colors $c_1, ..., c_i$ with $1 \leq i \leq j$.*

Note that Theorem 8 is a stronger formulation of Theorem 3 in the main text.

We first extend our definition of EC and $\text{EC}^m$ to a simplicial complex $K$ of dimension $n$ with a vertex coloring function $c : V(K) \to X$.

**Definition 8.** *Let $K$ be a simplicial complex of dimension $n$ with vertex color set $X$. For $0 < i \leq n$, we define a color value function $f_i : \prod_{j=1}^{i+1} X \to \mathbb{R}_{>0}$ as follows:*

1. *For any permutation $\sigma \in S_{i+1}$, $f_i(\sigma(\vec{x})) = f_i(\vec{x})$.*

2. *For any $\vec{x}, \vec{y} \in \prod_{j=1}^{i+1} X$ such that $\vec{x}$ and $\vec{y}$ have the same coordinates modulo order and multiplicity, $f_i(\vec{x}) = f_i(\vec{y})$. For example, if $X = \{red, blue\}$, $f_2(red, red, blue) = f_2(blue, blue, red)$. The intuition is, as sets (so we forget about order and multiplicity), $\{red, red, blue\}$ is the same set as $\{blue, blue, red\}$.*

*We define the $i$-th simplex filtration function $f_i$ of $K$ as follows:*

- *For all simplices $\sigma$ with dimension less than $i$, $F_i(\sigma) = 0$.*

- *For all $i$-simplices $\sigma$ with vertices $v_0, ..., v_i$, $F_i(\sigma) := f_i(c(v_0), ..., c(v_i))$.*

- *For all simplices $\sigma$ with dimension greater than $i$, $F_i(\sigma) := \max_{i\text{-simplex } \tau \subset \sigma} f_i(\tau)$.*

*Note that when $K = G$ is a graph $f_0, f_1, F_0, F_1$ agrees with our construction of $f_v, f_e, F_v, F_e$. Finally, for each $i$ and $t \in \mathbb{R}$, we define*

$$K_t^{f_i} := F_i^{-1}((-\infty, t]).$$

From here, we can construct the EC and max EC diagram on simplicial complexes as follows.

**Definition 9.** *Let $K$ be a simplicial complex of dimension $n$ with simplex filtration functions $f_0, f_1, ..., f_n$. The EC diagram $\text{EC}(K, f_0, ..., f_n)$ of $K$ is composed of $n + 1$ lists $\text{EC}(K, f_i)$ for $i = 0, ..., n + 1$. For each $i$, write $a_1^i < ... < a_{n_i}^i$ as the list of values $f_i$ can produce, $\text{EC}(K, f_i)$ is the list $\{\chi(K_{a_j^i}^{f_i})\}_{j=1}^{n_i}$.*

*Suppose we are only given a vertex filtration function $f_0$. We define the max EC diagram of $K$ as $\text{EC}^m(K, f_0) = \text{EC}(K, f_0, g_1, ..., g_n)$. Here $g_i(\sigma) = \max_{i\text{-simplex } \tau \subset \sigma} \max_{v_i \in \tau} f_0(v_i)$.*

One can check that the $g_i$'s defining max EC are consistent with the setup in Definition 8. As an immediate corollary of Theorem 8, we will see that EC and $\text{EC}^m$ have the same expressive power on the level of simplicial complexes.

**Theorem 9.** EC *and* $\text{EC}^m$ *have the same expressive power.*

**Remark 2.** *The choice of Condition (2) in Definition 8 is intentional so that Theorem 9 would hold. The intuition is that Condition (2) corresponds to coloring the $i$-th simplex $\sigma$, with vertices $v_0, ..., v_{i+1}$, of $K$ with the feature being the set $\{v_0, ..., v_{i+1}\}$ that forgets about multiplicity and order. However, Theorem 9 is not true if we modify Condition (2) such that we color $\sigma$ with the feature being the multi-set $\{c(v_0), ..., c(v_{i+1})\}$ to remember multiplicity.*

We will now prove Theorem 8. To do this, we will first prove a lemma.

**Lemma 1.** *Let $K$ and $M$ be two simplicial complexes of dimension $n$, if they have the same max EC diagram for all possible $f_0$, then $K$ and $M$ have the same $f$-vector, meaning that their respective number of simplicies are the same at each dimension.*

*Proof.* For this proof, let $K^i$ denote the subcomplex of $K$ with all simplicies with dimension $\leq i$. By comparing any complete filtration, we will have that $\chi(K) = \chi(M)$ (so $\chi(K^{n-1}) = \chi(M^{n-1})$). Comnparing time $t = 0$ for any induced $i$-simplex coloring filtration by an injective $f_0$ will give us that $\chi(K^i) = \chi(M^i)$ for all $i = 0, ..., n-1$. From here we can see that they have the same $f$-vector. $\qquad\square$

Now we will finally prove Theorem 8.

*Proof of Theorem 8.* Clearly (1) implies (2). To see that (3) implies (1), we first note that (3) implies $K$ and $M$ have the same $f$-vector, so the starting values of their Euler characteristics for any $f_i$ are always the same. For any change of the Euler characteristic as time varies, we will be modifying the value by adding or subtracting values of the form $S_K^j(c_1, ..., c_i) = S_M^j(c_1, ..., c_i)$. Hence, their Euler characteristics will agree as time varies.

For (2) implies (3), we first note that Lemma 1 implies $K$ and $M$ have the same $f$-vector. We will now prove the claim from a backward induction on $j = 0, 1, ..., n$. Indeed, when $j = n$, we have that

1. We will induct on $1 \leq i \leq n$. For $i = 1$, we wish to show that $S_K^n(c_1) = S_M^n(c_1)$ for all $c_1 \in C$. Note that $S_K^n(c_1)$ is just the number of $n$-simplices whose vertices all have color $c_1$. Choose $f_0$ such that $c_1$ has the minimum value, then under the induced max $n$-simplex coloring filtration, we have that

$$\chi(K_{f_0(c_1)}^{f_n}) = \chi(M_{f_0(c_1)}^{f_n})$$

Since $K$ and $M$ have the same $f$-vector by Lemma 1, we conclude that $S_K^n(c_1) = S_M^n(c_1)$.

2. Suppose this is true up to $1 \leq i = k < n$. We wish to show this is true for $i = k + 1$. Indeed, let $c_1, ..., c_{k+1}$ be $k + 1$ distinct colors. Choose $f_0$ such that $c_1$ has the minimum value, $c_2$ is the second smallest, and so on until $c_{k+1}$. Under the induced max $n$-simplex coloring filtration, we have that

$$\chi(K_{f_0(c_{k+1})}^{f_n}) = \chi(M_{f_0(c_{k+1})}^{f_n})$$

Subtracting their Euler characteristics at the time step $f_0(c_k)$, we have that

$$(-1)^n \sum_{a \in \mathcal{A}} S_K^n(a) = (-1)^n \sum_{a \in \mathcal{A}} S_M^n(a).$$

where $\mathcal{A}$ is the collection of subsets of $\{c_1, ..., c_{k+1}\}$ that contains $c_{k+1}$. Since the inductive hypothesis is true up to $i = k$, we know that for all $a \in \mathcal{A}$ such that $|a| \leq k$, $S_K^n(a) = S_M^n(a)$. Hence cancelling both sides gives us

$$\sum_{a \in \mathcal{A}, |a| > k} S_K^n(a) = \sum_{a \in \mathcal{A}, |a| > k} S_M^n(a).$$

Now, there is only one element in $\mathcal{A}$ with $|a| > k$, namely $a = \{c_1, ..., c_{k+1}\}$. Hence, we have proven this for the case $i = k + 1$.

3. Thus, by induction, we have shown this for $j = n$.

Now we will induct down from $j = n$. Indeed, suppose this is true up to $0 < j = k + 1 \leq n$, we wish to show this is true for $j = k$. Now we wish to show that $S_K^k(c_1, ..., c_i) = S_M^k(c_1, ..., c_i)$ for all distinct colors $c_1, ..., c_i$ and $1 \leq i \leq k$. We will do this by an induction on $i$.

1. For $i = 1$, we wish to show that $S_K^k(c_1) = S_M^k(c_1)$. Indeed, choose $f_0$ such that $c_1$ has the minimum value. Then we have that

$$\chi(K_{f_0(c_1)}^{f_k}) = \chi(M_{f_0(c_1)}^{f_k})$$

Since $K$ and $M$ have the same $f$-vector, cancelling that out gives us that

$$\sum_{\ell=k}^{n} S_K^\ell(c_1) = \sum_{\ell=k}^{n} S_M^\ell(c_1).$$

By our inductive hypotehsis on $j$, we know that $S_K^\ell(c_1) = S_M^\ell(c_1)$ for all $\ell > k$, hence subtracting them off gives us that $S_K^k(c_1) = S_M^k(c_1)$.

2. Suppose this is true up to $1 \leq i = \ell < k$. We wish to show this is true for $i = \ell + 1$. Indeed, let $c_1, ..., c_{\ell+1}$ be $\ell + 1$ distinct colors. Choose $f_0$ such that $c_1$ has the minimum value, $c_2$ is the second smallest, and so on until $c_{\ell+1}$. Under the induced max $k$-simplex coloring filtration, we have that

$$\chi(K_{f_0(c_{\ell+1})}^{f_k}) = \chi(M_{f_0(c_{\ell+1})}^{f_k})$$

Since they have the same Euler characteristics at the time step $f_0(c_{\ell+1})$, we can cancel those terms out and obtain

$$\sum_{a \in \mathcal{A}} \sum_{p=k}^{n} (-1)^p S_K^p(a) = \sum_{a \in \mathcal{A}} \sum_{p=k}^{n} (-1)^p S_M^p(a).$$

Here $\mathcal{A}$ is the collection of all subsets of $\{c_1, ..., c_{\ell+1}\}$ that contains the color $c_{\ell+1}$. Now by the inductive hypothesis on $j$, we know that $S_K^p(a) = S_M^p(a)$ for all $p > k$, so we can cancel the expressions and obtain

$$(-1)^p \sum_{a \in \mathcal{A}} S_K^k(a) = (-1)^p \sum_{a \in \mathcal{A}} S_M^k(a).$$

Now by the inductive hypothesis for $i$, we have that for all $|a| < \ell + 1$, $S_M^k(a) = S_K^k(a)$, hence we have that

$$(-1)^p \sum_{a \in \mathcal{A}, |a| > \ell} S_K^k(a) = (-1)^p \sum_{a \in \mathcal{A}, |a| > \ell} S_M^k(a).$$

There is only one subset of size $\ell + 1$, namely $\{c_1, ..., c_{\ell+1}\}$. Hence, we conclude that $S_K^k(c_1, ..., c_{\ell+1}) = S_M^k(c_1, ..., c_{\ell+1})$.

Thus, by the principle of induction, we have proven that (2) implies (3). $\qquad\square$

# E    Implementation details

## E.1    Datasets

With the exception of ZINC, all datasets originate from the TUDataset repository — a comprehensive benchmark suite commonly employed for assessing graph kernel methods and GNNs. The datasets can be accessed at https://chrsmrrs.github.io/datasets/docs/datasets/. ZINC(12K) corresponds to a subset of the widely used ZINC-250K collection of chemical compounds [28], which is particularly suited for molecular property prediction [12].

BREC is a benchmark designed to evaluate the expressiveness of graph neural networks (GNNs). It comprises 800 non-isomorphic graphs organized into 400 pairs, grouped into four categories: Basic, Regular, Extension, and CFI. The Basic category includes 60 pairs of 1-WL-indistinguishable graphs, while the Regular category contains 140 pairs of regular graphs further divided into simple regular, strongly regular, 4-vertex-condition, and distance-regular graphs. For a detailed description of the remaining categories and graph constructions, we refer the reader to Wang and Zhang [49].

For completeness, we also considered minimal Cayley graphs. These datasets have been used to assess the expressivity of graph models [2] and can be found at https://houseofgraphs.org/meta-directory/minimal-cayley .

Table 4 reports summary statistics of the real-world datasets used in this paper.

Table 4: Statistics of the datasets.

| Dataset | #graphs | #classes | Avg #nodes | Avg #edges |
|---|---|---|---|---|
| NCI1 | 4110 | 2 | 29.87 | 32.30 |
| PROTEINS (full) | 1113 | 2 | 39.06 | 72.82 |
| DHRF | 756 | 2 | 42.43 | 44.54 |
| NCI109 | 4127 | 2 | 29.68 | 32.13 |
| COX2 | 467 | 2 | 41.22 | 43.45 |
| ZINC | 12000 | - | 23.16 | 49.83 |

## E.2  Expressivity experiments

For each collection of minimal Cayley graphs with $n$ nodes (denoted by c-$n$), we compute all possible pairwise combinations. The BREC benchmark provides pairs of non-isomorphic graphs for each type: basic, regular, and extension — for computational reasons, we do not report results for CFI graphs. For both benchmarks, accuracy is measured as the fraction of pairs of graphs that can be distinguished by their 0-dim and 1-dim persistence diagrams.

For edge closeness, we obtain an edge-level filtration by computing node closeness centrality on the line graph of each input graph. Recall that the closeness centrality of a node $u$ is the reciprocal of its average shortest-path distance to all other nodes in the graph. We compute this quantity using `nx.closeness_centrality` from the `NetworkX` library [20].

We also consider the Jaccard index, defined for an edge $(u, v)$ as

$$J(u, v) = \frac{\#\mathcal{N}(u) \cap \mathcal{N}(v)}{\#\mathcal{N}(u) \cup \mathcal{N}(v)},$$

where $\mathcal{N}(u)$ denotes the set of neighbors of $u$ in the graph. We use $J(u, v)$ as an edge-level filtration value, computed via `nx.jaccard_coefficient`.

Finally, we employ the square clustering coefficient as a vertex-level filtration, which measures the fraction of possible squares (i.e., 4-cycles) incident to a vertex in the graph. We compute it using `nx.square_clustering`.

**Implementation note.**  For the sake of transparency, while preparing the final version of the manuscript, we found an error in an earlier implementation of our expressivity experiments that erroneously reported perfect accuracy for GProd. After fixing the bug, we observed that, with our initial (simple) filtration choices, GProd and standard PH mostly achieve on par performance. To better highlight the benefits of GProd, we revisited the experimental setup by considering stronger filtration functions and additional datasets (including minimal Cayley graphs). Importantly, the corrected results corroborate our overarching theme that considering PH on graph products contain richer information than the individual components. Further details are provided in the official github repo.

## E.3  Runtime experiments

The runtime experiments were conducted on Google Colab and written in Python. This was done in the standard Python 3 Google Compute Engine backend without any subscription. For the experiments comparing Theorem 4 with gudhi and union-find, we applied a vertex-based degree filtration on the pair of graphs $G$ and $H$ and compute the induced product filtration on $G \square H$. For the experiments comparing Theorem 5, we applied an edge-based filtration that is a modified version of the Forman-Ricci Curvature [44] for graphs. More precisely, for an edge $(u, w) \in G$, we apply

$$f_e(u, w) := 4 - \#\mathcal{N}(u) - \#\mathcal{N}(w) + 3\#(\mathcal{N}(u) \cap \mathcal{N}(w)) + 2\#V(G),$$

where $\#\mathcal{N}(\bullet)$ is the number of adjacent nodes and $\#V(G)$ is the number of nodes in the entire graph. This is just done to ensure $f_e > 0$.

## E.4  Experiments on graph classification

We implement all models using the PyTorch Geometric Library [17]. For all experiments, we use Tesla V100 GPU cards and consider a memory budget of 64GB of RAM.

For the TU datasets, we use a random 80/10/10% (train/validation/test) split, which varies with the random seed (i.e., a different split is used for each of the five runs). All models are trained with an initial learning rate of $10^{-3}$, which is halved if the validation accuracy does not improve for 10 consecutive epochs. We employ early stopping with a patience of 40 epochs and train for up to 500 epochs using the Adam optimizer [31]. A batch size of 64 and batch normalization are used in all experiments.

We apply the vectorization scheme of [8] with Gaussian point transformations, identity weight function, and mean aggregation. Formally, let $\mathcal{D} = \{p_1, \ldots, p_n\}$ be a persistence diagram with $n$ tuples $p_i \in \mathbb{R}^2$. We compute

$$h(\mathcal{D}) = \frac{1}{n} \sum_{i=1}^{n} \varphi(p_i), \tag{1}$$

where

$$\varphi(p) = [\Gamma_p(\phi_1), ..., \Gamma_p(\phi_q)]^\top \quad \text{with} \quad \Gamma_p(\phi_i) = \exp\left(-\frac{\|\phi_i - p\|_2^2}{2}\right), \tag{2}$$

and $\phi_1, \ldots, \phi_q \in \mathbb{R}^2$ are learnable parameters. In all experiments, we restrict our analysis to 0-dimensional persistence diagrams and adopt $q = 100$. Two filtration functions are considered: degree centrality and betweenness centrality. The optimal one is determined, together with the number of GNN layers, based on validation performance — i.e., the filtration function is a hyperparameter.

Here, the topological embeddings $h(\cdot)$ are concatenated with the graph-level GNN embeddings (obtained via a mean readout) and passed through an MLP consisting of two hidden layers with ReLU activations and a hidden dimension of 64.

We consider seven graphs as references in our approach, implemented in the `NetworkX` library: `nx.cycle_graph(4)`, `nx.star_graph(5)`, `nx.turan_graph(5,2)`, `nx.complete_graph(5)`, `nx.lollipop_graph(3,3)`, `nx.watts_strogatz_graph(10,2,0.2)`, and `nx.house_graph`.

