# OpenReview forum: "On topological descriptors for graph products"
_NeurIPS.cc/2025/Conference — NeurIPS 2025 poster_

### Official Review · Reviewer_quwK · 2025-06-29

**Clarity:** 3
**Significance:** 3
**Originality:** 3
**Rating:** 5
**Confidence:** 3

**Summary:**

This paper studies persistent homology (PH) and Euler characteristics (EC) of graph products and present several new theoretical results. The authors introduce a computationally cheaper variant of EC, called max-EC, and proves that it has the same expressive power for graphs and fore higher-dimensional simplicial complexes. It is shown that PH on graph products captures strictly more structural information than PH on individual graphs, while EC does not capture more information on graph products. The authors present new algorithms for computing PH persistence pairs (0 and 1 dimensional) for vertex/edge-level product filtrations, and new characterizations of product filtrations via colorings on graph products. As application, the authors propose to use graph product filtrations as enhanced topological descriptors in GNN.

**Questions:**

- The proposed algorithms for PH on product filtrations track births-deaths inductively, which seems expensive to scale. Do the authors has any estimation of their computational complexity for large graphs?

- The authors suggest that graph products can better capture relational data structures. Could the authors give some experiments or discuss some use cases where graph product aware PH improves GNN performance? Will the improvement outweigh the cost of product filtration computations?

- While using virtual nodes can boost PH expressivity, does it change or add some artificial features (like circles or cliques) to the graphs? If it does, how do the additional features affect the descriptors? How to choose f_v and f_e?

**Ethical Concerns:**

["NO or VERY MINOR ethics concerns only"]

**Final Justification:**

The original theoretical results are new and interesting and constitute a nice contribution to topological data analysis. During the rebuttal phase, the authors provide new results on complexity analysis and experimental results. They promise to revise the paper to incorporate new results provided during the rebuttal. Overall, the theoretical results are valuable contribution which justifies my rating of the paper.

**Limitations:**

Yes

**Quality:**

3

**Strengths And Weaknesses:**

Strengths:
- To my knowledge, max-EC and the virtual node technique for boosting PH expressivity in graph products are new.  max-EC gives a lightweight, theoretically grounded alterative to EC. Virtual nodes have been used to improve expressivity of GNN, but its use in TDA seems new.
- The combinatorial characterization of EC expressivity for graphs and simplicial complexes, as well as the study of PH/EC on graph products are highly valuable to topological data analysis. max-EC and PH might be useful for exploiting graph product structures in GNN.
- The proofs (edge and vertex color counting arguments, filtration refinements, Betti number calculations) look sound to me. The authors acknowledge that their algorithms for persistence pairs only apply to product filtrations, not general color-based filtrations.

 Weaknesses:
- It lacks discussions on computational complexity and scalability of the proposed algorithms for PH on product filtrations.

- It lacks experiments or concrete use cases where graph product aware PH improves GNN performance.

- It lacks discussion on the effects of virtual nodes on additional graph features and the appropriate choice of f_v and f_e.

---

> ### Author Rebuttal · Authors · 2025-07-31
>
> Thank you for your review and suggestions. We reply to your comments/questions below.
>
> ```
> It lacks discussions on computational complexity and scalability of the proposed algorithms for PH on product filtrations. ... Do the authors has any estimation of their computational complexity for large graphs?
> ```
> Thank you for this question! We have now implemented **Theorem 4** and **Theorem 5** in Python, and we will discuss their theoretical complexity and empirical runtime here.
>
> For a connected graph with m edges, its PH can be calculated in O(mlog(m)) time (see the discussion before 4.1 in [1]). If the graph is not connected, it is O(n + mlog(m)) where n is the number of vertices - to account for the worst-case scenario where the graph is totally disconnected.
>
> Now, let G be a graph with nG nodes, mG edges, H be graph with nH nodes, mH edges, and X be the coloring set for G and H.
>
> For **Theorem 5**, the algorithm does not need to compute the graph product and is dependent on the PH for G and PH for H separately. We can precompute the number of non-trivial deaths and number of vertices still alive in O(nG log(nG)) and O(nH log(nH)) time by sorting the two PH diagrams by death-times and count accordingly. Then we loop through both lists in parallel to compute the diagram. Thus, the runtime of Theorem 5 would be $O(max(nG log(nG)+nH log(nH), mG log(mG) + nG, mH log(mH) + nH))$.
>
> For **Theorem 4**, the algorithm also does not need to compute the product of the two graphs. We can similarly compute the two individual persistence diagrams in $O(mG log(mG) + nG)$ and $O(mH log(mH) + nH)$ time. We can pre-compute the relevant data from the diagrams similarly. If we assume X has constant size, then this can in fact be done in linear time. We initialize the persistence diagram with nG nH tuples with empty entries. In this case, when we loop through the filtration steps, our current (unoptimized) code implements the intermediate steps in the Theorem by looping through the nG*nH tuples. This yields the runtime as O(nG nH + mG log mG + mH log mH + nG + nH).
>
> In contrast, a naive calculation for vertex-level and edge-level pproduct filtrations would be to run PH on the graph product directly. This incurs a runtime of O(nG nH + (nG mH + nH mG) log(nG mH + nH mG)).
>
> **Theorem 4** and **Theorem 5** empirically perform significantly better than the naive implementations.
>
> For the runtime experiments of vertex-level product filtration, we considered three algorithms - one based on **Theorem 4**, one performing a naive calculation on the entire graph product using the union-find PH code outlined in Algorithm 1 of [2], and one performing a naive calculation using the **gudhi** library's PH code. We note that the **gudhi** library's PH code is very optimized and industry standard. We used the BREC dataset [3] for the data. For each dataset we considered, we load in two graphs G, H each time. We assign them vertex labels being the degree (ie. fv = deg) as the color and then compute the PH of G \Box H.
>
> The following table reports the average runtime (in seconds, taken over 10 trials) each algorithm took to go over the entire dataset.
> | **Method**       | **Basic**|**STR** | **DR** | **CFI**|
> |-------------------|--------|--------|----------|-------|
> |Theorem 4 (ours)| **0.967** |**2.040**|**1.554**|**31.253** |
> |Naive PD|1.427|33.691|22.569|673.253|
> |gudhi|1.091|16.469|10.243| 106.463|
>
> For experiments of edge-level product filtration, we considered again **Theorem 5**, naive PD with union find, and gudhi. The edge filtration function is given by an augmented Forman–Ricci curvatu with $f_e(u, w) = 4 - |N(u)| - |N(w)| + 3|N(u) \cap N(w)| + 2 n$, where $N(u)$ is the neighboring vertices of $u$ and $n$ is the number of total vertices. The following table shows the average time (in seconds) taken over 10 trials.
> | **Method**       | **Basic**|**STR** | **DR** | **CFI**|
> |-------------------|--------|--------|----------|-------|
> |Theorem 5 (ours)|0.859|**1.527**|**1.055**|**5.899**|
> |Naive PD|1.465|38.828| 24.138 |742.611|
> |gudhi| **0.065**|4.110| 2.179 |32.111|
>
> Finally, we analyze the theoretical costs of **Proposition 8**. For A and B graphs, since the quantity b1(A \Box B) can be expressed in terms of the number of edges (1), the number of connected components (2), and the number of vertices (3) of A and B respectively. To compute the 1-dim persistence diagram of a product filtration, it suffices for us to keep track of how (1), (2), (3) changes over time. These can all be done in a time dominated by O(mG log mG + nG + mH log mH + nH) (ie. the time to compute the persistence diagrams), which still outperforms the naive method.
>
> ```
> It lacks experiments or concrete use cases where graph product aware PH improves GNN performance. [...] Could the authors give some experiments or discuss some use cases where graph product aware PH improves GNN performance?
> ```
> Thank you for your comment. First, we emphasize that the proposed descriptors capture structural information that GNNs (1-WL) fail to detect. In particular, our topological descriptors distinguish all pairs of graphs in the BREC dataset [3], which is specifically designed to assess the expressivity of graph models.
>
> The accuracy results (fraction of identified non-isomorphic graph pairs) are:
> | **Filtration**   | **Diagram**       | **Reg** | **Ext** | **Basic** | **DR** | **STR** |
> |------------------|-------------------|--------:|--------:|----------:|-------:|-------:|
> | **Degree**       | PH$^V$ |    0.00 |    0.07 |      0.03 |   0.00 |   0.00 |
> |                  | RePHINE           |    0.94 |    0.77 |      0.97 |   0.00 |   0.00 |
> |                  | GProd$^V$ (ours)  |    1.00 |    1.00 |      1.00 |   1.00 |   1.00 |
> | **Betweenness**  | PH$^V$ |    0.96 |    0.82 |      0.95 |   0.00 |   0.00 |
> |                  | RePHINE           |    0.98 |    0.94 |      1.00 |   0.00 |   0.00 |
> |                  | GProd$^V$ (ours)  |    1.00 |    1.00 |      1.00 |   1.00 |   1.00 |
> | **Closeness**    | PH$^V$ |    0.26 |    0.46 |      0.27 |   0.00 |   0.00 |
> |                  | RePHINE           |    0.98 |    0.88 |      1.00 |   0.00 |   0.00 |
> |                  | GProd$^V$ (ours)  |    1.00 |    1.00 |      1.00 |   1.00 |   1.00 |
> | **FR Curvature** | PH$^E$ |    0.94 |    0.70 |      0.97 |   0.00 |   0.00 |
> |                  | GProd$^E$ (ours)  |    1.00 |    1.00 |      1.00 |   1.00 |   1.00 |
>
> Regarding real-world experiments, we consider 4 classification tasks and 2 GNNs: GIN and GraphSAGE. The results (over 5 seeds) are in the Table below.
>
>
> | Model                   | PROTEINS   | NCI1      | NCI109    | ZINC      |
> |-------------------------|------------|-----------|-----------|-----------|
> | **GIN**                 | 68.75 ± 1.55 | 80.62 ± 0.37 | 76.67 ± 1.61 | 0.58 ± 0.01 |
> | **GIN + ProdFiltration**| 69.05 ± 2.25 | 81.43 ± 0.51 | 77.72 ± 1.37 | 0.58 ± 0.01 |
> | **SAGE**                | 69.35 ± 2.25 | 77.62 ± 3.05 | 77.80 ± 1.38 | 0.54 ± 0.01 |
> | **SAGE + ProdFiltration**| 70.24 ± 2.87 | 78.02 ± 2.03 | 78.29 ± 1.46 | 0.52 ± 0.01 |
>
>
> We refer to our answer to Reviewer MD6t for further details on the experiments.
>
> ```
> Will the improvement outweigh the cost of product filtration computations?
> ```
> We note that combining product filtrations with GNNs scales very efficiently. Because these filtrations can be pre-computed, they introduce only a minor overhead to the standard GNN pipeline. Recall that, in product filtrations, we do not compute the Cartesian product between the graphs.
>
> In particular, the theoretical and empirical computational costs of the computational algorithms outlined in Section 5 are quite promising. Please see our response to the first weakness you pointed out above for more details.
>
> ```
> It lacks discussion on the effects of virtual nodes on additional graph features and the appropriate choice of f_v and f_e ... While using virtual nodes can boost PH expressivity, does it change or add some artificial features (like circles or cliques) to the graphs? If it does, how do the additional features affect the descriptors? How to choose f_v and f_e?
> ```
> Thank you for raising the question. For our case, we adjoin a virtual node to a graph G as a disjoint basepoint that is born at time negative infinity (ie. at the very start), write G_+ as G adjoined the virtual node. Note that this is different from what some other people might call ``virtual" (such as adding a fully adjacent node to connect to the entire graph, ie. [4]). In our case, adding a disjoint vertex to G does not increase the number of circles or cliques (unless you allow isolated vertices to be cliques in the definition), but it will add an extra component. The effect of having a virtual node is more significant for graph products. For both G and H, G_+ \Box H_+ is exactly the disjoint union of four graphs: an isolated node, a copy of G, a copy of H, and G \Box H. This is done to guarantee that a filtration of G_+ \Box H_+ also goes through G and H individually (which is possible because we also assign the virtual a distinct color away from the existing coloring set). To go through G and H first, one should choose fv and fe such that they spawn G and H first in the series (or at least in an isolated time interval).
>
> ----------
> Many thanks for your insightful review and constructive feedback. We hope your concerns have been satisfactorily addressed, and if so, would appreciate if you could revisit your score to reflect the same. We are also committed to engaging further if you have any additional questions, concerns, or suggestions.
>
> [1] Horn et al, Topological graph neural networks,ICLR 2022.
>
> [2] Immonen et al, Going beyond persistent homology using persistent homology.NeurIPS 2024.
>
> [3] Wang et al, Towards Better Evaluation of GNN Expressiveness with BREC Dataset,ICLR 2024.
>
> [4] Alon et al, On the Bottleneck of Graph Neural Networks and its Practical Implications,ICLR 2021.

---

> > ### Comment · Reviewer_quwK · 2025-08-03
> > **Rebuttal discussion**
> >
> > I thank the authors for the rebuttal. My overall judgement of this submission is negatively impacted by the large amount of additional results, cost estimation and many new experimental results. Compared to the original submission, which is supposed to be a complete work for reviewing, significant new results are added in the rebuttal on (1) Details on cost analysis and Python codes with experiments for Theorems 4 and 5, (2) Many important experimental results in various setting to show the advantages of the proposed method GraphProd.
> >
> > The authors promise to revise and update the manuscript, but the task of reviewers is to review the submitted version, not a significantly modified manuscript. In a sense, the submitted paper is premature for submission which I find concerning. I have some further questions for the discussion:
> >
> > Question 1: Could the authors detail their plan to revise the paper with new results? Will the revised version be significantly different from the submitted version? It seems useful to know how the revised version looks like.
> >
> > Question 2: There are some previous works on learning product spaces which is missing the references, for example: Zhang et al., "Switch Spaces: Learning Product Spaces with Sparse Gating", https://arxiv.org/abs/2102.08688. As the narrative of the paper is now led partially to showing advantages with many experimental results, how will the author plan to discuss this line of related works and compare the performance between capturing graph product structures using persistent homology and learning it automatically?

---

> > > ### Author Response · Authors · 2025-08-03
> > >
> > > Thank you for your continued engagement. We address your questions below.
> > > ```
> > > Plans for revision
> > > ```
> > > Thank you for the opportunity to clarify this. We believe it is our obligation to allay concerns and questions raised by all reviewers during the rebuttal, including questions explicitly on computation cost and experiments. As we describe below, **we do not need to significantly alter the original material, while still being able to accommodate the key questions raised by you and all the other reviewers in the main text.**. We have in fact added the following changes to the manuscript and verified that the NeurIPS main page length is respected, while all the original results listed in Figure 1 remain as is.
> > >
> > > 1. We add a 1-sentence statement about the respective cost analysis at the end of Theorem 4, 5, and Prop 8, For example, at the end of Theorem 4: "The runtime is $O(n_Gn_H+m_G\log m_G+m_H \log m_H+n_G+n_H)$" right after the sentence "The remaining vertices after $a_n$ all die at $\infty$."
> > >
> > > The detailed analysis of runtimes is accordingly added to the proofs of these theorems in the Appendix.
> > >
> > > 2. We add a new section after Section 5 to give an overview and discuss the experiments. As is standard these days, we provide the details on the datasets and implementation in the Appendix.
> > >
> > > 3. To account for the additional space due to these additions, we move some rather auxiliary/secondary details to the Appendix; specifically,
> > >
> > > - The paragraph before Prop 8, which explains it suffices to look at the first Betti number of the graph product when computing 1-dimensional persistence; this paragraph was directly referenced in the proof of Prop 8 in the appendix of the original submission anyway.
> > >
> > > - Example 1 and its Fig 2 (which gives an example for computing EC diagram). This minor restructuring is reasonable since it does not break the flow in the main paper that still contains the method and all the key results, and the readers curious about the details of the computation can check out the example in the Appendix.
> > >
> > > - Example 3 and its Fig 5 (which gives an example using Theorem 5 as an algorithm).
> > >
> > > The statements of Theorem 4,5 share similarities in how they are structured (they both involve refining a filtration step into 2 steps and computing intermediate quantities). Theorem 5 is also simpler than Theorem 4 since we only need to keep track of the death time of the vertices as they were all born at 0,but in Theorem 4 we need to pay attention to both the birth and death times.
> > >
> > > Thus, we believe it is ok to move Example 3 and Fig 5 to the Appendix while keeping Example 2 and Fig 4 in the main paper. Again, this slight restructuring is reasonable since Theorem 5 shares thematic similarities with Theorem 4 as mentioned above.
> > >
> > > In summary, the presentation of key aspects of all results in the original submission remains unaffected. Including the computational cost and the experiments adds only just over a page of content in the main paper, which as we explained above can be handled seamlessly.
> > >
> > > ```
> > > There are some previous works on learning product spaces which is missing ... how will the author plan to discuss ... related works and compare the performance between capturing graph product structures using persistent homology and learning it automatically
> > > ```
> > > Thank you for the question. Although we both have the word "product", the paper referenced (Zhang et al.) is not really connected to our work. The paper by Zhang et al. is about selecting the top K manifolds out of a product of manifolds with varying dimensions and curvatures. Our work, in contrast, investigates how PH and EC behave on a product of graphs. A graph is almost never a manifold. For any graph that has a vertex v of degree > 2, v does not have a locally Euclidean neighborhood and hence G is not a manifold.
> > >
> > > Moreover, the product in their paper is quite different from our product operation. The product in Zhang et al is about the topological product of spaces. In our work, we work with the box product of graphs. A box product of graphs is not the topological product because the topological product of two 1-dim objects would be 2-dim (ex. the topological product of two path graphs would be a filled square), whereas the box product of two graphs is still a graph. Zhang et al. also never discussed persistent homology.
> > >
> > > As far as we know, we are not aware of any other prior work in GNNs that combines the box product of graphs with topological descriptors. If there are any other papers you would like us to take a look at, we would of course be happy to.
> > >
> > > ---
> > > We’re grateful for this discussion that has allowed us to elaborate on some salient aspects of this work, and restructure some content to ensure that the constructive suggestions by you and the other reviewers are reflected in the revised version, without having to decrease the technical depth, conceptual foundations, and the key contributions of the original submission. Thank you so much!

---

> > > > ### Comment · Reviewer_quwK · 2025-08-03
> > > > **Rebuttal response**
> > > >
> > > > I thank the authors for their detailed response. My concerns with whether adding so many new results will significantly change the submitted manuscript are not fully resolved, as I think the complexity analysis and experimental results are of significant importance to this work. So briefly adding it to a minor revision of the paper will not be very satisfying. My question on comparison between feature engineering vs end to end learning of product graph structure was misinterpreted by the authors, but I won't discuss this point further, as I realize now that it may not directly related to topological data analysis. I think the initial theoretical results are new and interesting, and hope that the authors will improve their work into a complete, throughout study. I will keep my score as 5.

---

> > > > > ### Author Response · Authors · 2025-08-04
> > > > >
> > > > > Thank you for your feedback and for recognizing the novelty and interest of our initial theoretical results! The questions on computational complexity and experiments have surely improved our work. We are committed to developing the work further, and thank you for maintaining your score.

---

### Official Review · Reviewer_6YNL · 2025-07-02

**Clarity:** 3
**Significance:** 3
**Originality:** 3
**Rating:** 4
**Confidence:** 4

**Summary:**

This paper investigates topological descriptors on graph products,  EC and PH. Specifically, the authors prove the expressive power of EC and PH on graph products and propose a PH computation algorithm tailored for filtrations on such structures.

**Questions:**

The authors could at least propose some preliminary experimental designs, which would significantly strengthen the paper. For example, one could verify that graph products enhance the expressive power of persistent homology (PH) on synthetic graphs [1]. Additionally, for a graph classification task, it would be valuable to compare the effectiveness of PH and Euler characteristic (EC) under the graph product structure, observing the performance improvements of classification models [2].

[1] Ballester R, Rieck B. On the expressivity of persistent homology in graph learning[J]. arXiv preprint arXiv:2302.09826, 2023.

[2] Zhao Q, Ye Z, Chen C, et al. Persistence enhanced graph neural network[C]//International Conference on Artificial Intelligence and Statistics. PMLR, 2020: 2896-2906.

**Ethical Concerns:**

["NO or VERY MINOR ethics concerns only"]

**Final Justification:**

I am pleased to see that the authors have added experiments and verified the effectiveness of their method. I believe the authors have proposed an interesting method that is also valuable to the community, and recommending the acceptance of this paper is a natural choice.

**Limitations:**

The authors have already discussed the key limitations of their work.

**Paper Formatting Concerns:**

No formatting issues were identified.

**Quality:**

3

**Strengths And Weaknesses:**

**Strengths**

1. The study of topological data analysis methods on graph products is both novel and insightful. It has the potential to inspire future work in the machine learning community.

**Weaknesses**

1. As a submission to a machine learning conference, the absence of experiments is a major shortcoming. Although the authors emphasize the potential of PH in neural network applications, no experimental results are provided to support this claim.

2. The lack of an experimental section also affects the overall structure of the paper. In particular, Section 3 feels disconnected and could be moved to the appendix to allow the main text to focus more clearly on topological descriptors for graph products. Accordingly, the introduction should also be revised to reflect this structural shift.

---

> ### Author Rebuttal · Authors · 2025-07-31
>
> Thank you for your suggestions to strengthen our work.
>
> ### Experiments
> ```
> As a submission to a machine learning conference, the absence of experiments is a major shortcoming. Although the authors emphasize the potential of PH in neural network applications, no experimental results are provided to support this claim.
> ```
> Thanks for your comment. As suggested, we have added three sets of experiments:
> 1) expressivity on BREC datasets --- here, surprisingly, we found that simple filtration functions on Graph Products can distinguish all pairs of graphs on BREC.
>
> 2) graph classification on NCI1, NCI109, PROTEINS, ZINC. Here, we showed that the topological descriptors consistently boost the performance of GNNs (SAGE, GIN).
>
> 3) Experimental validations for the runtime benefits of the proposed algorithms in **Theorem 4** and **Theorem 5**.
>
> Please, see the next answers for details and results.
>
> ```
> The authors could at least propose some preliminary experimental designs, which would significantly strengthen the paper. For example, one could verify that graph products enhance the expressive power of persistent homology (PH) on synthetic graphs [1].
> ```
> The Table below shows the accuracy results of different topological descriptors in distinguishing pairs of graphs of the BREC benchmark [1]. We compare our proposal (GraphProd) against PH (vertex and edge filtrations) and RePHINE [3]. We consider different filtration functions: degree, betweenness centrality, closeness centrality and Forman-Ricci curvature. Surprisingly, GraphProd distinguishes all graphs, including the challenging ones: Distance Regular (DR) and strongly regular (STR), for which the other descriptors cannot distinguish any pair. This highlights the expressive power of using graph products for detecting graph isomorphism.
>
> | **Filtration**   | **Diagram**       | **Reg** | **Ext** | **Basic** | **DR** | **STR** |
> |------------------|-------------------|--------:|--------:|----------:|-------:|-------:|
> | **Degree**       | PH$^V$ |    0.00 |    0.07 |      0.03 |   0.00 |   0.00 |
> |                  | RePHINE           |    0.94 |    0.77 |      0.97 |   0.00 |   0.00 |
> |                  | GProd$^V$ (ours)  |    1.00 |    1.00 |      1.00 |   1.00 |   1.00 |
> | **Betweenness**  | PH$^V$ |    0.96 |    0.82 |      0.95 |   0.00 |   0.00 |
> |                  | RePHINE           |    0.98 |    0.94 |      1.00 |   0.00 |   0.00 |
> |                  | GProd$^V$ (ours)  |    1.00 |    1.00 |      1.00 |   1.00 |   1.00 |
> | **Closeness**    | PH$^V$ |    0.26 |    0.46 |      0.27 |   0.00 |   0.00 |
> |                  | RePHINE           |    0.98 |    0.88 |      1.00 |   0.00 |   0.00 |
> |                  | GProd$^V$ (ours)  |    1.00 |    1.00 |      1.00 |   1.00 |   1.00 |
> | **FR Curvature** | PH$^E$ |    0.94 |    0.70 |      0.97 |   0.00 |   0.00 |
> |                  | GProd$^E$ (ours)  |    1.00 |    1.00 |      1.00 |   1.00 |   1.00 |
>
>
>
> ```
> Additionally, for a graph classification task, it would be valuable to compare the effectiveness of PH and Euler characteristic (EC) under the graph product structure, observing the performance improvements of classification models [2].
> ```
> To assess the effectiveness of product filtration (Theorem 4) in real-world classification tasks, we can employ classifiers as in the following:
>
> 1. **Define a family of small reference graphs**. We selected eight graphs generated using the NetworkX library: nx.cycle_graph(4),nx.complete_graph(5),nx.star_graph(5), nx.lollipop_graph(3,3),nx.watts_strogatz_graph(10,2,0.2),nx.turan_graph(5,2), nx.house_graph();
>
> 2. **Compute pairwise product filtrations**. For an incoming graph G, we compute the product filtration (using betweenness centrality as filtration function) between G and each reference graph. This will result in 8 persistence diagrams.
>
> 3. **Vectorize the diagrams**. We vectorize the obtained diagrams from previous step using any suitable scheme (e.g., DeepSets or the methods in [2]). In this work, we applied Gaussian vectorization [2] with q=10
>
> 4. **Merge topological and GNN embeddings**. We concatenate the topological embeddings with GNNs embeddings before sending it through a classification head.
>
>
> We are interested in assessing if these topological descriptors based on graph products can improve the performance of GNNs. We considered 2 GNN models: GIN and GraphSAGE with either 2, 3 or 4 layers (selected via validation set). We assessed the models in 4 real-world datasets: NCI1, NCI109, ZINC, and PROTEINS. Here are the results (over 5 seeds):
>
> | Model                   | PROTEINS   | NCI1      | NCI109    | ZINC      |
> |-------------------------|------------|-----------|-----------|-----------|
> | **GIN**                 | 68.75 ± 1.55 | 80.62 ± 0.37 | 76.67 ± 1.61 | 0.58 ± 0.01 |
> | **GIN + ProdFiltration**| 69.05 ± 2.25 | 81.43 ± 0.51 | 77.72 ± 1.37 | 0.58 ± 0.01 |
> | **SAGE**                | 69.35 ± 2.25 | 77.62 ± 3.05 | 77.80 ± 1.38 | 0.54 ± 0.01 |
> | **SAGE + ProdFiltration**| 70.24 ± 2.87 | 78.02 ± 2.03 | 78.29 ± 1.46 | 0.52 ± 0.01 |
>
> As we can see, the topological descriptors lead to boost across most combinations of GNNs and datasets. Overall, this illustrates the practical relevance of our contribution. We will add this to the revised manuscript.
>
> **Experimental Validations of Run-Times for Theorem 4 and 5:**
> For the runtime experiments of vertex-level product filtration, we considered three algorithms - one based on **Theorem 4**, one performing a naive calculation on the entire graph product using the union-find PH code outlined in Algorithm 1 of [2], and one performing a naive calculation using the **gudhi** library's PH code. We note that the **gudhi** library's PH code is very optimized and industry standard. We used the BREC dataset [3] for the data. For each dataset we considered, we load in two graphs G, H each time. We assign them vertex labels being the degree (ie. fv = deg) as the color and then compute the PH of G \Box H.
>
> The following table reports the average runtime (in seconds, taken over 10 trials) each algorithm took to go over the entire dataset.
> | **Method**       | **Basic**|**STR** | **DR** | **CFI**|
> |-------------------|--------|--------|----------|-------|
> |Theorem 4 (ours)| **0.967** |**2.040**|**1.554**|**31.253** |
> |Naive PD|1.427|33.691|22.569|673.253|
> |gudhi|1.091|16.469|10.243| 106.463|
>
> For experiments of edge-level product filtration, we considered again **Theorem 5**, naive PD with union find, and gudhi. The edge filtration function is given by an augmented Forman–Ricci curvatu with $f_e(u, w) = 4 - |N(u)| - |N(w)| + 3|N(u) \cap N(w)| + 2 n$, where $N(u)$ is the neighboring vertices of $u$ and $n_ is the number of total vertices. The following table shows the average time (in seconds) taken over 10 trials.
> | **Method**       | **Basic**|**STR** | **DR** | **CFI**|
> |-------------------|--------|--------|----------|-------|
> |Theorem 5 (ours)|0.859|**1.527**|**1.055**|**5.899**|
> |Naive PD|1.465|38.828| 24.138 |742.611|
> |gudhi| **0.065**|4.110| 2.179 |32.111|
>
> We can see that as the graph data gets larger, our **Theorem 4 and 5** significantly outperforms the other two methods here. We also analyzed the theoretical complexity of the two theorems. For more details on that, please see our response to **Reviewer quwK**.
>
> ### Section 3
> ```
> In particular, Section 3 feels disconnected and could be moved to the appendix to allow the main text to focus more clearly on topological descriptors for graph products.
> ```
> We agree as well - we will need space to add in the experiments.
>
> -----
> Many thanks for your insightful review and constructive feedback. We hope your concerns have been satisfactorily addressed, and if so, would appreciate if you could revisit your score to reflect the same. We are also committed to engaging further if you have any additional questions, concerns, or suggestions.
>
> [1] Towards Better Evaluation of GNN Expressiveness with BREC Dataset, ICLR 2024.
>
> [2] PersLay: A Neural Network Layer for Persistence Diagrams and New Graph Topological Signatures, AISTATS 2021.
>
> [3] Immonen et al, Going beyond persistent homology using persistent homology.NeurIPS 2024.

---

> > ### Comment · Reviewer_6YNL · 2025-08-01
> >
> > I am glad to see that the authors have constructed a variety of experiments to validate the effectiveness of their method. Based on this, I will raise my score. I also encourage the authors to include more representative baselines in the final version to better demonstrate the advantages of their approach. I look forward to seeing the improved manuscript.

---

> > > ### Author Response · Authors · 2025-08-04
> > >
> > > Thank you for your positive response! We are glad to hear that your concerns have been addressed, and we will for sure (and have) incorporate the experiments in the final version. Thank you for deciding to raise your score - we would appreciate it if this can be reflected in the original review as well.

---

### Official Review · Reviewer_ETh2 · 2025-07-03

**Clarity:** 2
**Significance:** 2
**Originality:** 3
**Rating:** 4
**Confidence:** 5

**Summary:**

The authors establish a characterization of the expressivity of persistent homology (PH) and Euler Characteristic (EC) of the filtrations over graph products. The authors also establish a characterization of EC on general color-based filtrations. Further, they show that PH of graph products captures strictly more information than PH of single graphs while EC does not. They also give algorithms to compute the PH diagrams of vertex- and edge-level filtrations on the graph product.

**Questions:**

1. In line 65, shouldn’t it be $E \subseteq V \times V$, instead of $E \subseteq V \times X$?

2. What is the larger picture of having defined coloring filtrations? One could have just started with two different filtration functions: first where one starts with a filtration function on the vertices and extends is piecewise linearly onto the edges, and another where all the vertices have zero filtration values and has some non-zero values on edges. I feel defining the coloring based filtrations, especially the way it is defined is rather confusing for the meaning it is conveying.

3. Line 91-92: What do you mean by “filtration of a diagram”?

4. The definition of $f_v$, seems to imply that one has a function $f_v$ for each vertex $v$. Moreover, since the function is not defined as $f_v : V \to \mathbb{R}$, but as $f_v : X \to \mathbb{R}$, it is understandable that each vertex has a potentially different function $f_v$. However, the writing seems to hint at the fact that $f_v$ is just one function that is defined as a filtration function on the vertices, which doesn’t seem right because the domain is not the set of vertices. Moreover, if it is so, then what is the point of the coloring function $c$? One could just consider $f_v \circ c$ as a filtration function on the vertices which is the standard setting for PH?

5. Moreover, it seems like this is an analysis of a bifiltration over a graph where $f_v$ and $f_e$ are two different filtrations.

**Ethical Concerns:**

["NO or VERY MINOR ethics concerns only"]

**Final Justification:**

The authors have engaged extensively in discussions and have clarified the questions I had. I believe that this method can be useful for the Topological Machine Learning community and would recommend a borderline accept.

**Limitations:**

Yes.

**Quality:**

2

**Strengths And Weaknesses:**

Strengths:

The paper is well-presented. I like the boxes around theorems.

Weaknesses:

The choice of notation for $F_v$ seems slightly confusing. I think having the notation depending on $f_v$ makes more sense than $v$. $f_v$ makes sense because it is tied to a vertex $v$. However, $F_v$ is not tied to a vertex, rather it is tied to the function $f_v$.

The authors have motivated the need for graph products in a few lines towards the beginning of Section 4. However, the motivation seems lacking in concrete intuition and examples for the need of graph products. E.g., "From an expressivity standpoint, graph products also capture information that the individual components may not have", this line is vague and doesn't get across the idea that the authors are trying to convey.

Refer to Questions.

Typos:

Line 65: “A graph is a the…” -> “A graph is the data…”?

---

> ### Author Rebuttal · Authors · 2025-07-31
>
> Thank you for your time and comments. We reply to your comments/questions below.
>
> ### Motivation
> ```
> The authors have motivated the need for graph products in a few lines towards the beginning of Section 4. However, the motivation seems lacking in concrete intuition and examples for the need of graph products. E.g., "From an expressivity standpoint, graph products also capture information that the individual components may not have", this line is vague and doesn't get across the idea that the authors are trying to convey.
> ```
> Thank you for raising this question! Here we give 2 motivations for graph products - 1. they show up naturally in the context of relational databases and 2. they can perform quite well in expressivity tests.
>
> **Relational Databases**:
> Recall that for vertices (g1, h1), (g2, h2) in G \Box H, we say there is an edge between them if either (g1 = g2 and h1 ~ h2 in H) or (g1 ~ g2 in G and h1 = h2).
>
> Suppose for example you have two tables - Table A is the wage income of residents in New York, and Table B is the wage income of residents in London, with some additional demographic / financial information for both. For each table, we view the rows as nodes. Now assign a similarity metric on the rows (ex. we are grouping people with similar income and background together), such that two nodes with sufficient similarity are linked with an edge.
>
> Now suppose your goal is to compare the residents in New York vs the residents in London based on this, then one natural operation would be to take the Cartesian product of the tables, call this new table "Table C". A reasonable graph one can make out of Table C is as follows - the nodes are still the rows. Let us think about how edges should be made. For person X in Table A paired with person Y in Table B, in order to analyze how X compares with Y and those similar to Y, we ought to look at how X connects with everyone in Table B that are connected to Y under the similarity metric before. What this means in terms of graphs is that we should link the node (X, Y) to (X, Y') for all Y' that is linked to Y. We can also do this symmetrically, and we can see that the construction of a graph under this heuristics is exactly the graph (box) product of Table A and B.
>
> Thus, we see that graph products show up naturally in the context of relational databases, and we will expand on this in the introduction.
>
> **Expressivity Tests**:
> Another motivation comes from that graph products appear to do quite well empirically in expressivity tests. We considered expressivity tests using the BREC benchmark [1]. In this regard, we show that Graph Products allow distinguishing all pairs of non-isomorphic graphs in BREC, surpassing previous topological descriptors (e.g., RePHINE [2]). The results (accuracy) are:
>
>
> | **Filtration**   | **Diagram**       | **Reg** | **Ext** | **Basic** | **DR** | **STR** |
> |------------------|-------------------|--------:|--------:|----------:|-------:|-------:|
> | **Degree**       | PH$^V$ |    0.00 |    0.07 |      0.03 |   0.00 |   0.00 |
> |                  | RePHINE           |    0.94 |    0.77 |      0.97 |   0.00 |   0.00 |
> |                  | GProd$^V$ (ours)  |    1.00 |    1.00 |      1.00 |   1.00 |   1.00 |
> | **Betweenness**  | PH$^V$ |    0.96 |    0.82 |      0.95 |   0.00 |   0.00 |
> |                  | RePHINE           |    0.98 |    0.94 |      1.00 |   0.00 |   0.00 |
> |                  | GProd$^V$ (ours)  |    1.00 |    1.00 |      1.00 |   1.00 |   1.00 |
> | **Closeness**    | PH$^V$ |    0.26 |    0.46 |      0.27 |   0.00 |   0.00 |
> |                  | RePHINE           |    0.98 |    0.88 |      1.00 |   0.00 |   0.00 |
> |                  | GProd$^V$ (ours)  |    1.00 |    1.00 |      1.00 |   1.00 |   1.00 |
> | **FR Curvature** | PH$^E$ |    0.94 |    0.70 |      0.97 |   0.00 |   0.00 |
> |                  | GProd$^E$ (ours)  |    1.00 |    1.00 |      1.00 |   1.00 |   1.00 |
>
>
> ### Typos
> ```
> Line 65: “A graph is a the…” -> “A graph is the data…”?
> ```
> Thanks for catching this. We will fix it in the revision.
>
> ```
> In line 65, shouldn’t it be E \subset V x V, instead of E \subset V x X
> ```
> Thanks for catching the typo. We will fix it in the revision.
> ```
> Line 91-92: What do you mean by “filtration of a diagram”?
> ```
> Thanks for catching the typo, we meant **graph** instead of **diagram**. We will fix it in the revision.
>
>
> ### Color Filtrations
> ```
> The choice of notation for Fv seems slightly confusing...
>
> What is the larger picture of having defined coloring filtrations? One could have just started with two different filtration functions: first where one starts with a filtration function on the vertices and extends is piecewise linearly onto the edges, and another where all the vertices have zero filtration values and has some non-zero values on edges. I feel defining the coloring based filtrations, especially the way it is defined is rather confusing for the meaning it is conveying.
>
> The definition of f_v seems to imply ...  the writing seems to hint at the fact that $f_v$ is just one function that is defined as a filtration function on the vertices, which doesn’t seem right because the domain is not the set of vertices.
> ```
> Thank you for the comments. To clarify, the lower script v in $f_v$ does not represent a vertex v on a graph. As explained in Definition 1, $f_v$ is defined with respect to a color set $X$ rather than a specifc vertex. We will clarify this and change the notation $f_v$ to $f_{vx}$ and $f_e$ to $f_{ed}$ if you think that would help with the clarity.
>
> $(f_v, f_e)$ is defined with respect to a coloring set $X$. The benefit of defining them this way is that, for any graph $G$ with a coloring set $X$, we can use the pair $(f_v, f_e)$ to produce an induced filtration of $G$. This is very important in terms of expressivity tests.
>
> For example, if we have two graphs $G, H$ with the same coloring set $X$. We want to use the PH of some filtration to determine if they are isomorphic or not. The issue is that, even if G and H are isomorphic, it is not in general true that a random filtration on G and a random filtration on H would give the same PH diagram. Therefore, randomly selecting filtrations on G and H respectively would not make a valid graph isomorphism test.
>
> Therefore, in order for the graph isomorphism test using PH to be well-defined, the filtration on $G$ and the filtration on $H$ must be dependent on one another in some way. This is to ensure that if G and H are isomorphic, the graph isomorphism tests here would not give false negatives. Now, since G and H have the same coloring set $X$, one way to find two filtrations that are related to each other is if they are both induced by the same pair $(f_v, f_e)$. In this case, the PH diagram would be an isomorphism invariant! (See [3])
>
> This is why we defined $(f_v, f_e)$ (and consequently the color-based filtrations) so that the correspondent graph isomorphism test with the PH diagrams would be valid. To not confuse $f_v$ with the actual filtration defined on the graph $G$, we use $F_v$ to denote essentially $f_v \circ c$ as in Definition 1.
>
> ```
> what is the point of the coloring function c? One could just consider f_v o c ...
> ```
> The point of having the color function $c$ is because we want the color set $X$ to be intrinsically defined (it is supposed to represent attributes labeled to the graph) as opposed to being a subset of the real numbers. If we have considered $f_v \circ c$ directly without passing through an intermediate $X$, this means that we are assigning real numbers as the attributes of the graph. However, often times the attributes of the graph may include non-numerical data. This is really a special case where the colr set $X$ is a subset of the real numbers. The purpose of having the function $c: V(G) \to X$ is that it would retain the specific attributes that the graph would have without ``converting" them into real numbers directly.
>
> We do remark that, in practice, sometimes we would assign the color on the vertex as a real number directly and take $f_v$ to be the identity, such as when we did the expressivity tests. But note that we could totally modify $f_v$ to get other kinds of filtrations while keeping the same colors.
>
> ```
> Moreover, it seems like this is an analysis of a bifiltration over a graph where fv and fe are two different filtrations.
> ```
> Thank you for the question. If by a **bifiltration**, you mean a **2-parameter** filtration, then the answer is no. In 2-parameter persistence, we define a function h: G -> R^2 (instead of h: G -> R). There is no time anymore, but rather a partial order on R^n. In our case, we can define a 2-parameter filtration of h: G -> R^2 as the collection G_{s,t} = h^{-1}( (-infty, s] x (-infty, t]) for (s,t) in R^2 (which also changes only finitely many times). The important observation to note is that there is now a matrix of graphs as opposed to a line of subgraphs in the 1-parameter case. This is different from looking at fv and fe separately in our case!
>
> ---
> Many thanks for your insightful review and constructive feedback. We hope your concerns have been satisfactorily addressed, and if so, would appreciate if you could revisit your score to reflect the same. We are also committed to engaging further if you have any additional questions, concerns, or suggestions.
>
> [1] Towards Better Evaluation of GNN Expressiveness with BREC Dataset, ICLR 2024.
>
> [2] Immonen et al, Going beyond persistent homology using persistent homology.NeurIPS 2024.
>
> [3] On the Expressivity of Persistent Homology in Graph Learning. Ballester and Rieck.

---

> > ### Comment · Reviewer_ETh2 · 2025-08-04
> >
> > I would like to begin by thanking the authors for their efforts!
> >
> > "For example, if we have two graphs $G, H$ with the same coloring set $X$. We want to use the PH of some filtration to determine if they are isomorphic or not. The issue is that, even if G and H are isomorphic, it is not in general true that a random filtration on G and a random filtration on H would give the same PH diagram. Therefore, randomly selecting filtrations on G and H respectively would not make a valid graph isomorphism test."
> >
> > Well, for testing for graph isomorphism, one can consider a simple vertex degree based filtration, which would be sufficient to test for graph isomorphism using PH. I am not fully convinced by this argument.
> >
> >
> > "The point of having the color function $c$ is because we want the color set $X$ to be intrinsically defined (it is supposed to represent attributes labeled to the graph) as opposed to being a subset of the real numbers. If we have considered $f_v \circ c$ directly without passing through an intermediate $X$, this means that we are assigning real numbers as the attributes of the graph. However, often times the attributes of the graph may include non-numerical data. This is really a special case where the colr set $X$ is a subset of the real numbers. The purpose of having the function $c: V(G) \to X$ is that it would retain the specific attributes that the graph would have without ``converting" them into real numbers directly.
> >
> > We do remark that, in practice, sometimes we would assign the color on the vertex as a real number directly and take $f_v$ to be the identity, such as when we did the expressivity tests. But note that we could totally modify $f_v$ to get other kinds of filtrations while keeping the same colors."
> >
> > I understand the point the authors are trying to make. However, I do not see the point in it because, eventually, even if you do retain information about specific attributes of the graph, computing PH is going to get the topological information into numerical data (either index-persistence or real-number persistence).
> >
> >
> > Thank you for clarifying the notation about color filtrations.
> >
> > With these points in mind, I would like to retain my score.

---

> ### Author Response · Authors · 2025-08-05
>
> Thank you for your continued engagement. We respond to your concerns/comments as follows.
>
> ```
> Well, for testing for graph isomorphism, one can consider a simple vertex degree based filtration, which would be sufficient to test for graph isomorphism using PH. I am not fully convinced by this argument.
> ```
> Thank you for the opportunity to clarify our work. The example you gave here can be seen as an example of assigning colors to unlabeled graphs. In this case, the colors on each vertex would be its degree, and the degree-based filtration you described would be to spawn the vertices according to their degrees from lowest to highest, and an edge is spawned if both of its vertices are spawned.
>
> However, this is only one way to spawn the vertices. We could have also spawned the vertices according to their degrees from highest to lowest. The perspective of **coloring filtration** is that we can spawn the vertices in any permutation of the set of possible degree values here. For example, if the set of degrees from two graphs G and H are {1, 3, 5, 8}. A typical degree-based filtration would be to spawn all the vertices of degree 1 first, then degree 3, then degree 5, and finally degree 8. The perspective of coloring filtration is to consider all possible permutations - ie. 1-3-5-8, 1-3-8-5, 3-1-5-8, 3-8-5-1, etc... - for which we can take PH.
>
> In practice, one sometimes cannot test all possible permutations, but doing the degrees sequentially in time is only 1 of the n! possibility. The perspective of **coloring filtration** is that one should not prefer a permutation over another. In general, there are pairs of non-isomorphic graphs such that taking the PH of their degree-based filtration (lowest to highest) cannot differentiate, but taking the PH of their degree-based filtration in a different order does. And this is only one of the other possible colors in general.
>
> For example, let G (left) and H (right) be the following 2 non-isomorphic graphs made in the networkX library
> ```
> G = nx.from_edgelist([(0, 1), (1, 2), (2, 3), (1, 4), (4, 5)])
> H = nx.from_edgelist([(0, 1), (1, 2), (1, 3), (3, 4), (4, 5)])
> ```
> In this case, the possible vertex degrees are 1, 2, 3. Running PH on the vertex degree-based filtration (the version you proposed) will not be able to differ them (this is the order 1-2-3), neither would doing this in the order 3-2-1. However, we can differentiate the 2 graphs in the order 2-3-1. Indeed, if we load all the vertices labeled the color/degree '2' first, we would get 2 connected components in G but 1 connected component in H. Thus, there are graphs that **a traditional degree filtration cannot differ but coloring filtrations can**.
>
> We also note that a vertex degree-based filtration did not manage to differentiate the majority of pairs of graphs in the BREC dataset (see our table in the initial rebuttal comment with PH$^V$). However, if we incorporate alternative colors in, it could potentially increase their expressivity over them.
>
> ```
> I understand the point the authors are trying to make. However, I do not see the point in it because, eventually, even if you do retain information about specific attributes of the graph, computing PH is going to get the topological information into numerical data (either index-persistence or real-number persistence).
> ```
> Thank you for your question. In light of the explanation above, we hope it makes more sense why we wanted attributes to be in X. This is because the **coloring filtration** considers all the possible permutations of colors rather than a single order. Having the colors stored numerically may lead to some confusion since R has a natural order.
>
> That being said, we also do not see any major issues with labeling the colors into numerical attributes to begin with, which is an example of assigning colors. In the perspective of **coloring filtrations**, they are just assigning ids to the vertices. The usual filtration in the order of numerical values is a special example of a coloring filtration where $f_v$ is the identity.
>
> For example, if we have a graph G where the set of numerical values are the vertices are {56, 90.4, 101.7}. We can spawn in all the vertices labeled 56 first, and then 90.4, and finally 101.7. Or we can down 90.4 first, and then 101.7, and finally 56. There are 6 possibilities in total here. The essence of **coloring filtrations** just asks to consider alternative orders to filtrate than the usual sequential one.
>
> ---
> Many thanks for your questions! We hope this addresses your concerns, and, if so, would appreciate if you could change your rating to reflect the same.
>
> [1] Towards Better Evaluation of GNN Expressiveness with BREC Dataset, ICLR 2024.

---

> > ### Comment · Reviewer_ETh2 · 2025-08-05
> >
> > I would like to thank the authors for their quick response.
> >
> > While I get the point they are trying to make in the example given, it is a non-example because the statement I made in my comment was for the case when G and H are isomorphic. In their original rebuttal, the authors say, "The issue is that, even if G and H are isomorphic, it is not in general true that a random filtration on G and a random filtration on H would give the same PH diagram. Therefore, randomly selecting filtrations on G and H respectively would not make a valid graph isomorphism test." I suggested a simple degree based filtration for the case when G and H are isomorphic. However, the example given is the case where the graphs are non-isomorphic to begin with and hence is a non-example.

---

> > > ### Author Response · Authors · 2025-08-05
> > >
> > > Thank you for your continued engagement. We respond to your comment as follows.
> > >
> > > ```
> > >  I suggested a simple degree based filtration for the case when G and H are isomorphic. However, the example given is the case where the graphs are non-isomorphic to begin with and hence is a non-example.
> > > ```
> > >
> > > Thank you for the opportunity to clarify our work. We first note that all the graph isomorphism tests we have been discussing (the simple degree-based filtration you are considering, and the coloring filtration we are considering) are not complete invariants. In other words, while they guarantee that isomorphic graphs have the same diagram (no false negatives), non-isomorphic graphs can have the same diagram (there are false positives). Therefore, the baseline for these tests is that the test should not create false negatives, and in general, we only have conclusive results if the functions return **False**. (Otherwise, it is unknown if checking two graphs are isomorphic completely can be done in polynomial time.)
> > >
> > > In this perspective, any method that returns a Boolean value and does not create false negatives on isomorphic graphs is a **graph isomorphism test**. The simple degree based filtration you proposed is one, and the coloring filtration we are considering is one. However, suppose you could just ask the function to always return **True** on a pair of graphs without any filtration (call this method **ReturnTrue**). This is also a valid **graph isomorphism test** because it always returns True for the case when G and H are isomorphic, so it creates no false negatives.
> > >
> > > If we ask you - would you prefer **ReturnTrue** or **Degree-Based PH** as a graph isomorphism test? While they both satisfy the condition you outlined in the comment you gave - "[they both return True] for the case when G and H are isomorphic". You might argue degree-based PH is better, because there are pairs of non-isomorphic graphs G and H that degree-based PH can differ but not **ReturnTrue**. While it is true that **ReturnTrue** always returns true when G and H are isomorphic (and so is **degree-based PH**), explaining why **Degree-Based PH** is better than **ReturnTrue** necessitates the need to look at non-isomorphic pairs of graphs.
> > >
> > > What we were trying to say in the last comment is a similar situation. By our previous discussion, **there are pairs of non-isomorphic graphs that degree-based PH cannot distinguish, but coloring-based PH can**. Furthermore, as we can see in the previous comment, **the expressive power of looking at the PH of a simple degree-based filtration is strictly bounded by the expressive power of its coloring filtrations**.
> > >
> > > While it is important for graph isomorphism tests to always return True on isomorphic graphs, this is the baseline for all graph isomorphism tests to be valid and only the beginning. To compare between different graph isomorphism tests, **expressivity** is one such quantity, in which case **coloring-based PH** outperforms **degree-based PH**. This is because **coloring filtration** is a generalization of the **degree-based filtration** you suggested, as assigning degrees to vertices can be seen as a special example of assigning colors, for which we can control the order for which they spawn by an ordering on the colors. In the case where the colors are ordered linearly as their real numbers assigned, this is the degree-based filtration you suggested. A **coloring filtration** is a natural generalization and hence contains more expressive power.
> > >
> > > If we only care about graph isomorphism tests on pairs of isomorphic graphs, then every graph isomorphism test proposed is no better than **ReturnTrue**. To obtain any meaningful comparisons, we have to look at pairs of non-isomorphic graphs to assess the abilities of the tests given.
> > >
> > > ---
> > > Many thanks for your questions! We hope this addresses your concerns, and, if so, would appreciate if you could change your rating to reflect the same.

---

> > > > ### Comment · Reviewer_ETh2 · 2025-08-06
> > > >
> > > > Yes I got the point that you were trying to make in your last answer itself. You want to establish an if and only if condition, which I understand.
> > > >
> > > > Thanks for engaging and answering all my questions. I have updated my score.

---

### Official Review · Reviewer_Ytkj · 2025-07-03

**Clarity:** 4
**Significance:** 3
**Originality:** 4
**Rating:** 5
**Confidence:** 4

**Summary:**

This paper analyzes topological descriptors (persistent homology (PH) and Euler characteristic (EC)) and their behavior when lifting to (Cartesian) graph products, providing a thorough theoretical framework for understanding their behavior under color-based filtrations.

Key contributions include:

- The introduction of max PH and max EC diagrams, with proofs of bijections between vertex and edge-level filtrations.
- A complete characterization of EC expressivity for graphs with color filtrations, showing EC and its proposed “max” variant are equally expressive yet the max version is more computationally efficient.
- A formal study of **graph products**, demonstrating that while EC gains no new expressivity on products, PH of product graphs (with  the addition of “virtual” nodes) is strictly more expressive than for components alone.
- Algorithms to compute **0- and 1-dimensional persistence diagrams** for product filtrations derived from vertex- and edge-level colorings, showing how the product filtration can be decomposed into simpler component filtrations.

**Questions:**

1. Can you more clearly motivate why product graphs matter for ML tasks? E.g., subgraph GNN architectures, relational data modeling—concrete scenarios or prior work where your theory would plug in? Here I think elaborating in particular on the connection to relational databases modeling via GNNs and how this relates to product graphs could take a front seat in the introduction. I think EC and PH are well-studied and motivated, but the need for the calculus and understanding these descriptors when lifting to products would be better placed here than in Section 4.
2. Could you separate out and clarify the definition of the **max-induced edge coloring function**? Currently it's embedded in the EC diagram definition and I think it might be more clear (including in the reference in Prop 1) if this got its own definition and explanation and set up the motivation for why this produces the "max EC diagram" and collapses the need to deal with both vertex and edge coloring functions.

**Ethical Concerns:**

["NO or VERY MINOR ethics concerns only"]

**Final Justification:**

The updated experiments and improved motivation significantly strengthen the paper’s impact and clarity, addressing key concerns and justifying my increased score.

**Limitations:**

yes

**Quality:**

3

**Strengths And Weaknesses:**

**Strengths:**

- **Technical depth and rigor:** The paper presents formal, well-structured theorems (e.g., Theorem 2's equivalence between EC and max EC) with detailed, convincing proofs.

- **Novelty:** The approach to EC expressivity via vertex-based filtrations and their “max” counterpart for edge-based filtrations offers a compelling unification, highlighting an instructive example where PH achieves greater expressivity than EC—an interesting trade-off given EC’s computational efficiency.

- **Potential impact:** By formalizing product filtrations and developing a clear calculus for them, the paper provides a solid foundation for improved topological descriptors on graph products. This framework can exploit decomposition properties and clarifies contexts where PH’s expressivity has clear advantages over EC.

**Weaknesses:**

- **Motivation for ML community is underdeveloped:** While the paper mentions subgraph GNNs and multi-way data, the introduction lacks concrete use cases or experiments demonstrating practical relevance. For example, the connection to relational databases (Robinson et al.) is super interesting and could be motivated further to captivate a more general audience.
- **Justification:** Solid theoretical contribution with meaningful advances in understanding topological descriptors on graph products. It's a borderline accept because its practical ML motivation needs strengthening. I'd be open to raising my score if the authors clarified motivation and improved exposition.

---

> ### Author Rebuttal · Authors · 2025-07-31
>
> We are grateful for your time and comments. We reply to your comments/questions below.
>
> ```
> Motivation for ML community is underdeveloped: While the paper mentions subgraph GNNs and multi-way data, the introduction lacks concrete use cases or experiments demonstrating practical relevance.
> ```
> Thanks for your comment. We conducted two sets of experiments to demonstrate the practical relevance of our paper.
> First, we consider expressivity tests using the BREC benchmark [1]. In this regard, we show that Graph Products allow distinguishing all pairs of non-isomorphic graphs in BREC, surpassing previous topological descriptors (e.g., RePHINE). The results (accuracy) are:
>
>
> | **Filtration**   | **Diagram**       | **Reg** | **Ext** | **Basic** | **DR** | **STR** |
> |------------------|-------------------|--------:|--------:|----------:|-------:|-------:|
> | **Degree**       | PH$^V$ |    0.00 |    0.07 |      0.03 |   0.00 |   0.00 |
> |                  | RePHINE           |    0.94 |    0.77 |      0.97 |   0.00 |   0.00 |
> |                  | GProd$^V$ (ours)  |    1.00 |    1.00 |      1.00 |   1.00 |   1.00 |
> | **Betweenness**  | PH$^V$ |    0.96 |    0.82 |      0.95 |   0.00 |   0.00 |
> |                  | RePHINE           |    0.98 |    0.94 |      1.00 |   0.00 |   0.00 |
> |                  | GProd$^V$ (ours)  |    1.00 |    1.00 |      1.00 |   1.00 |   1.00 |
> | **Closeness**    | PH$^V$ |    0.26 |    0.46 |      0.27 |   0.00 |   0.00 |
> |                  | RePHINE           |    0.98 |    0.88 |      1.00 |   0.00 |   0.00 |
> |                  | GProd$^V$ (ours)  |    1.00 |    1.00 |      1.00 |   1.00 |   1.00 |
> | **FR Curvature** | PH$^E$ |    0.94 |    0.70 |      0.97 |   0.00 |   0.00 |
> |                  | GProd$^E$ (ours)  |    1.00 |    1.00 |      1.00 |   1.00 |   1.00 |
>
>
>
> In addition, we demonstrate how the proposed descriptors can be combined with GNNs to improve predictive performance on four graph-classification datasets. Our strategy is as follows:
>
> 1. **Define a family of small reference graphs**. We selected eight graphs generated using the NetworkX library: nx.cycle_graph(4), nx.complete_graph(5), nx.star_graph(5), nx.lollipop_graph(3,3), nx.watts_strogatz_graph(10,2,0.2), nx.turan_graph(5,2), nx.house_graph();
>
> 2. **Compute pairwise product filtrations**. For an incoming graph G, we compute the product filtration (using betweenness centrality as filtration function) between G and each reference graph. This will result in 8 persistence diagrams.
>
> 3. **Vectorize the diagrams**. We vectorize the obtained diagrams from previous step using any suitable scheme (e.g., DeepSets or the methods in [2]). In this work, we applied Gaussian vectorization [2] with q=10
>
> 4. **Merge topological and GNN embeddings**. We concatenate the topological embeddings with GNNs embeddings before sending it through a classification head.
>
> We consider GINs (graph isomorphism networks) and GraphSAGE. The results are:
>
> | Model                   | PROTEINS   | NCI1      | NCI109    | ZINC      |
> |-------------------------|------------|-----------|-----------|-----------|
> | **GIN**                 | 68.75 ± 1.55 | 80.62 ± 0.37 | 76.67 ± 1.61 | 0.58 ± 0.01 |
> | **GIN + ProdFiltration**| 69.05 ± 2.25 | 81.43 ± 0.51 | 77.72 ± 1.37 | 0.58 ± 0.01 |
> | **SAGE**                | 69.35 ± 2.25 | 77.62 ± 3.05 | 77.80 ± 1.38 | 0.54 ± 0.01 |
> | **SAGE + ProdFiltration**| 70.24 ± 2.87 | 78.02 ± 2.03 | 78.29 ± 1.46 | 0.52 ± 0.01 |
>
> As we can see, the topological descriptors consistently yield improvements across most GNN–dataset combinations, underscoring the practical significance of our contribution.
>
> [1] Towards Better Evaluation of GNN Expressiveness with BREC Dataset, ICLR 2024.
>
> [2] PersLay: A Neural Network Layer for Persistence Diagrams and New Graph Topological Signatures, AISTATS 2021.
>
> ```
> Justification: Solid theoretical contribution with meaningful advances in understanding topological descriptors on graph products. It's a borderline accept because its practical ML motivation needs strengthening. I'd be open to raising my score if the authors clarified motivation and improved exposition.
> ...
> Can you more clearly motivate why product graphs matter for ML tasks? E.g., subgraph GNN architectures, relational data modeling—concrete scenarios or prior work where your theory would plug in? Here I think elaborating in particular on the connection to relational databases modeling via GNNs and how this relates to product graphs could take a front seat in the introduction. I think EC and PH are well-studied and motivated, but the need for the calculus and understanding these descriptors when lifting to products would be better placed here than in Section 4.
> ```
> Thank you for raising this question! Here, we elaborate on why graph products show up in terms of relational databases.
>
> Recall that for vertices (g1, h1), (g2, h2) in G \Box H, we say there is an edge between them if either (g1 = g2 ^ h1 ~ h2 in H) or (g1 ~ g2 in G and h1 = h2).
>
> Suppose for example you have two tables - Table A is the wage income of residents in New York, and Table B is the wage income of residents in London, with some additional demographic / financial information for both. For each table, we view the rows as nodes. Now assign a similarity metric on the rows (ex. we are grouping people with similar income and background together), such that two nodes with sufficient similarity are linked with an edge.
>
> Now suppose your goal is to compare the residents in New York vs the residents in London based on this, then one natural operation would be to take the Cartesian product of the tables, call this new table "Table C". A reasonable graph one can make out of Table C is as follows - the nodes are still the rows. Let us think about how edges should be made. For person X in Table A paired with person Y in Table B, in order to analyze how X compares with Y and those similar to Y, we ought to look at how X connects with everyone in Table B that are connected to Y under the similarity metric before. What this means in terms of graphs is that we should link the node (X, Y) to (X, Y') for all Y' that is linked to Y. We can also do this symmetrically, and we can see that the construction of a graph under this heuristics is exactly the graph (box) product of Table A and B.
>
> Thus, we see that graph products show up naturally in the context of relational databases, and we will expand on this in the introduction.
>
> We now also provide experimental evidence to illustrate the merits of filtrations on graph products in terms of expressivity benefits. Furthermore, such filtrations can also be used for performing graph classification. Please see the responses above for further details.tails.
>
> ```
> Could you separate out and clarify the definition of the max-induced edge coloring function? Currently it's embedded in the EC diagram definition and I think it might be more clear (including in the reference in Prop 1) if this got its own definition and explanation and set up the motivation for why this produces the "max EC diagram" and collapses the need to deal with both vertex and edge coloring functions.
> ```
> Thank you for the question. To clarify, a color-based filtration in Def 1 requires a pair of functions fv and fe. In a max-induced filtration, the function fe is dependent on the function fv. In particular, for an edge (a, b), fe(a, b) := max(fv(a), fv(b)) (ie. it takes the value of the vertex with greater value under fv). By a max-induced filtration, we mean color-based filtration in Def 1 with some fv and fe defined in terms of fv as above. max EC is the restriction of EC to only the max-induced filtrations. We will clarify this in the paper.
>
> **Additional Notes:** Also, we wanted to let you know that we analyzed the theoretical run times of **Theorem 4** and **Theorem 5** as well, and we did empirical experiments to demonstrate the benefits of their run-times against two naive implementations - one computing the PH of the whole graph product using an union-find method for PH, and another computing the PH of the whole graph using the industry standard PH calculation function from gudhi. For both theorems, when the size of the graph scales up, they perform significantly better than the two naive methods.
>
> The following two tables report the average runtime (in seconds, taken over 10 trials) each algorithm took to go over the entire dataset (taken from BREC).
>
>  | **Method**       | **Basic**|**STR** | **DR** | **CFI**|
> |-------------------|--------|--------|----------|-------|
> | Theorem 4 (ours) | **0.967** |**2.040**|  **1.554** | **31.253** |
> | Naive PD | 1.427  | 33.691  |  22.569 | 673.253 |
> | gudhi  |  1.091  |16.469| 10.243 | 106.463 |
>
> | **Method**       | **Basic**|**STR** | **DR** | **CFI**|
> |-------------------|--------|--------|----------|-------|
> | Theorem 5 (ours) | 0.859 | **1.527** | **1.055**  |**5.899**|
> | Naive PD | 1.465  |38.828| 24.138 |742.611|
> | gudhi  | **0.065**|4.110| 2.179 |32.111|
>
> Please see more details in the response to **Reviewer quwK**.
>
> ----
> Many thanks for your insightful review and constructive feedback. We hope your concerns have been satisfactorily addressed, and if so, would appreciate if you could revisit your score to reflect the same. We are also committed to engaging further if you have any additional questions, concerns, or suggestions.

---

> > ### Comment · Reviewer_Ytkj · 2025-08-01
> >
> > Thank you for the comprehensive response and the detailed experiments addressing all reviewer suggestions. I’m excited to see these improvements incorporated into the final draft. With the additions you’ve described, I believe this work will be a strong, impactful, and accessible contribution to the machine learning community—reflected in my updated score.

---

> > > ### Author Response · Authors · 2025-08-03
> > >
> > > Thank you for your positive response! We are glad to hear that your concerns have been addressed, and we will for sure incorporate your helpful feedbacks and the discussions here in the final version.

---

### Official Review · Reviewer_MD6t · 2025-07-03

**Clarity:** 3
**Significance:** 3
**Originality:** 3
**Rating:** 4
**Confidence:** 3

**Summary:**

This submission provides a theoretical analysis of different topological descriptors, i.e., Euler characteristics (EC) and persistent homology (PH). Based on the definition of the EC diagram and max EC diagram for EC, and box product for graph product, it analyzes the expressiveness power of EC and PH and show that PH gains strictly more information on graph products than the individual components. Two algorithms are provided for computing PH diagrams by focusing on the product of vertex-level and edge-level filtrations.

**Questions:**

1. What is the possible application of the proposed TDs, can you provide some experimental results for verification?
2. Can we use the proposed descriptors to improve the traditional GNN expressiveness? Or how to use them for practical tasks?
3. Besides the expressiveness, do we need to consider other metrics and aspects to assess / evaluate the topological descriptors? if yes, how about the results on them for the proposed EC and PH descriptors?
4. The author discuss the persistent descriptors of a graph, what is the counterpart? Nonpersistent? what is the advantage of considering the persistent TD rather than the others?

**Ethical Concerns:**

["NO or VERY MINOR ethics concerns only"]

**Final Justification:**

Most of my concerns are addressed based on the response provided by the author. However, the clarification and concise of the paper, especially the W1, should be taken seriously and hope they can be improved.

**Quality:**

3

**Strengths And Weaknesses:**

Strengths:
   - The paper focuses on the theoretical treatment of topological descriptors based on the color-based filtration and provides rigorous proofs and clear characterizations of their properties from the expressiveness perspective; it also provides algorithm frameworks for computing the PH diagrams.

Weakness:
   - The definition and theoretical formulation are dense, the provided example are not sufficiently explained, which may cause some difficulty for following for people without similar background.
   - There is no applications for the provided TD and experimental verification for the expressiveness power of defined diagrams and proposed algorithms, which also make the practical value of the proposed TD and algorithms unclear.
   - It only focuses on the color-based filtration on graph and lacks comprehensive comparison with other types of filtrations and expressiveness aspects;  the analysis of the proposed algorithms in Theorem 4 and 5 are insufficient, e.g., computation complexity and computational cost.

---

> ### Author Rebuttal · Authors · 2025-07-31
>
> Thank you for your review - we reply below.
>
> ```
> ... theoretical formulation are dense, ... may cause some difficulty for following for people without similar background.
> ```
> Thanks for pointing this out. We will revise the background section to add more clarity. We can also expand more on additional background in the appendix if you think it would be good. In part to aid the ease of understanding for people who come from a less theoretical background, we have now implemented both **Theorem 4 and 5** in executable Python codes with experiments around them. Seeing the actual code should be much more parseable, and we will also adjust the descriptions in the theorems to look more like pseudocodes.
>
> ```
> There is no applications for the provided TD and experimental verification for the expressiveness power of defined diagrams and proposed algorithms, which also make the practical value of the proposed TD and algorithms unclear. [...] can you provide some experimental results for verification?
> ```
> Thanks for your comment. We evaluate expressivity using the BREC datasets~\cite{brec}.
>
> Given a pair of graphs $G$ and $H$, we apply persistent homology (PH)
> on graph products to decide graph isomorphism as follows:
>
>   1. Compute the persistence diagrams for $G \Box G$ and $G \Box H$.
>     If the diagrams differ, we conclude that $G$ and $H$ are
>     **not isomorphic**. Otherwise, proceed to Step~2.
>
>   2. Compute the persistence diagrams for $G \Box G$ and $H \Box H$.
>     If these diagrams differ, we again conclude that $G$ and $H$ are
>     not isomorphic. If they are identical, the test is deemed
>     inconclusive (i.e., $G$ and $H$ are considered isomorphic by this test).
>
> Following this strategy, we compare our method (GraphProd) against PH (with vertex and edge filtrations) and RePHINE, considering several filtration functions: degree, betweenness centrality, closeness centrality, and Forman–Ricci curvature. Remarkably, GraphProd distinguishes all graphs, including the challenging Distance Regular (DR) and strongly regular (STR) cases, where the other descriptors fail to differentiate any pair.
>
> | **Filtration**   | **Diagram**       | **Reg** | **Ext** | **Basic** | **DR** | **STR** |
> |------------------|-------------------|--------:|--------:|----------:|-------:|-------:|
> | **Degree**       | PH$^V$ |    0.00 |    0.07 |      0.03 |   0.00 |   0.00 |
> |                  | RePHINE           |    0.94 |    0.77 |      0.97 |   0.00 |   0.00 |
> |                  | GProd$^V$ (ours)  |    1.00 |    1.00 |      1.00 |   1.00 |   1.00 |
> | **Betweenness**  | PH$^V$ |    0.96 |    0.82 |      0.95 |   0.00 |   0.00 |
> |                  | RePHINE           |    0.98 |    0.94 |      1.00 |   0.00 |   0.00 |
> |                  | GProd$^V$ (ours)  |    1.00 |    1.00 |      1.00 |   1.00 |   1.00 |
> | **Closeness**    | PH$^V$ |    0.26 |    0.46 |      0.27 |   0.00 |   0.00 |
> |                  | RePHINE           |    0.98 |    0.88 |      1.00 |   0.00 |   0.00 |
> |                  | GProd$^V$ (ours)  |    1.00 |    1.00 |      1.00 |   1.00 |   1.00 |
> | **FR Curvature** | PH$^E$ |    0.94 |    0.70 |      0.97 |   0.00 |   0.00 |
> |                  | GProd$^E$ (ours)  |    1.00 |    1.00 |      1.00 |   1.00 |   1.00 |
>
> [1] Towards Better Evaluation of GNN Expressiveness with BREC Dataset, ICLR 2024.
>
> [2] PersLay: A Neural Network Layer for Persistence Diagrams and New Graph Topological Signatures, AISTATS 2021.
>
> ```
> It only focuses on the color-based filtration on graph and lacks comprehensive comparison with other types of filtrations and expressiveness aspects; the analysis of the proposed algorithms in Theorem 4 and 5 are insufficient, e.g., computation complexity and computational cost.
> ```
> Please see the response above for experiments using filtraitons based on degree, betweeness, closeness, and FR curvature. We have implemented **Theorem 4** and **Theorem 5** in Python, their theoretical complexity and empirical runtime are discussed in more details in the response to **Reviewer quwk**.
>
> Here we summarize the main points.
>
> 1. For **Theorem 5**, the runtime is $O(max(nG log(nG)+nH log(nH), mG log(mG) + nG, mH log(mH) + nH))$.
>
> 2. Assuming the list of colors X is O(1), then for **Theorem 4**, the runtime is $ O(nG nH + mG log mG + mH log mH + nG + nH)$.
>
> 3. We did empirical experiments to demonstrate the benefits of their run-times against two naive implementations - one computing the PH of the whole graph product using an union-find method for PH, and another computing the PH of the whole graph using the industry standard PH calculation function from gudhi. For both theorems, when the size of the graph scales up, they perform significantly better than the two naive methods.
>
> 4. The following two tables report the average runtime (in seconds, taken over 10 trials) each algorithm took to go over the entire dataset (taken from BREC).
>
>  | **Method**       | **Basic**|**STR** | **DR** | **CFI**|
> |-------------------|--------|--------|----------|-------|
> | Theorem 4 (ours) | **0.967** |**2.040**|  **1.554** | **31.253** |
> | Naive PD | 1.427  | 33.691  |  22.569 | 673.253 |
> | gudhi  |  1.091  |16.469| 10.243 | 106.463 |
>
> | **Method**       | **Basic**|**STR** | **DR** | **CFI**|
> |-------------------|--------|--------|----------|-------|
> | Theorem 5 (ours) | 0.859 | **1.527** | **1.055**  |**5.899**|
> | Naive PD | 1.465  |38.828| 24.138 |742.611|
> | gudhi  | **0.065**|4.110| 2.179 |32.111|
>
> ```
> Can we use the proposed descriptors to improve the traditional GNN expressiveness? Or how to use them for practical tasks?
> ```
> Thanks to your question, we run experiments for graph classification on real-world data to demonstrate how the proposed descriptors can be used to boost the performance of GNNs. For this, we employ the following procedure:
>
> 1. **Define a family of small reference graphs**. We selected eight graphs generated using the NetworkX library: nx.cycle_graph(4),nx.complete_graph(5),nx.star_graph(5), nx.lollipop_graph(3,3),nx.watts_strogatz_graph(10,2,0.2),nx.turan_graph(5,2), nx.house_graph();
>
> 2. **Compute pairwise product filtrations**. For an incoming graph G, we compute the product filtration (using betweenness centrality as filtration function) between G and each reference graph. This will result in 8 persistence diagrams.
>
> 3. **Vectorize the diagrams**. We vectorize the obtained diagrams from previous step using any suitable scheme (e.g., DeepSets or the methods in [2]). In this work, we applied Gaussian vectorization [2] with q=10
>
> 4. **Merge topological and GNN embeddings**. We concatenate the topological embeddings with GNNs embeddings before sending it through a classification head.
>
> We considered 2 GNN models: GIN and GraphSAGE with either 2, 3 or 4 layers (selected via validation set). We assessed the models in 4 real-world datasets: NCI1, NCI109, ZINC, and PROTEINS. Here are the results:
>
> | Model                   | PROTEINS   | NCI1      | NCI109    | ZINC      |
> |-------------------------|------------|-----------|-----------|-----------|
> | **GIN**                 | 68.75 ± 1.55 | 80.62 ± 0.37 | 76.67 ± 1.61 | 0.58 ± 0.01 |
> | **GIN + ProdFiltration**| 69.05 ± 2.25 | 81.43 ± 0.51 | 77.72 ± 1.37 | 0.58 ± 0.01 |
> | **SAGE**                | 69.35 ± 2.25 | 77.62 ± 3.05 | 77.80 ± 1.38 | 0.54 ± 0.01 |
> | **SAGE + ProdFiltration**| 70.24 ± 2.87 | 78.02 ± 2.03 | 78.29 ± 1.46 | 0.52 ± 0.01 |
>
> As we can see, the topological descriptors lead to boost across most combinations of GNNs and datasets. Overall, this illustrates the practical relevance of our contribution.
>
>
> ```
> Do we need to consider other metrics and aspects to assess / evaluate the topological descriptors? if yes, how about the results on them for the proposed EC and PH descriptors?
> ```
> Computational costs and scalability are also important metrics. Please see our response to the third weakness above for more details on how they interact with PH. EC itself is much simpler and faster to compute due to being less expressive.
>
> We also made an implementation of **EC** and **max EC** in Python. To run a small experiment, we took the first 10 pairs of graph from basic.npy in the BREC dataset and used **EC** and **max EC** to run expressivity tests. For each pair of graphs G and H, we pre-assign them with vertex colors using the degree function. We then iterate through the list of possible color-based filtrations until the two diagrams either differ or we exhaust the list.
>
> In a single pass, **max EC** finished the tests in 0.311 seconds. **EC** finished the tests in 69.624 seconds. The reason why is because of **Theorem 1** - for **max EC**, we only need to look at a sub-collection of **max-induced filtration** to get the same power, so the code only needed to only go through that sub-collection for **max EC**
>
> ```
> The author discuss the persistent descriptors of a graph, what is the counterpart? Nonpersistent? what is the advantage of considering the persistent TD rather than the others?
> ```
> The counterpart would be a **static** descriptor. A **persistent** descriptor for a graph is a method that involves some computations with respect to a filtration of the graph. Persistent homology, for example, looks at how connected components and cycles are born and killed in a given filtration of a graph, so these descriptors typically track an evolution of some information over time. A static descriptor does not look at a filtration, it only looks like the graph in its final state and produces some output. For example, the number of connected components and cycles of the whole graph G is a static descriptor, whereas reporting how these two information changes over time would be a persistent descriptor. One reason why persistent descriptors may be more preferable than static ones is because it is often more expressive. This is because the static one only looks at what happens at the very end, whereas persistent looks at what happens in the filtration of the graph.

---

> > ### Author Response · Authors · 2025-08-09
> > **Rebuttal Follow-Up**
> >
> > Dear Reviewer MD6t,
> >
> > As the discussion period is ending soon, we wanted to check-in again. We have responded to your concerns in the rebuttal, notably:
> > - We conducted experiments around the **expressivity, run-time efficiency, and GNN classification tasks** that our theory could do, showing that each case leads to empirical gains.
> > - In our experiments, we considered various forms of filtrations given by **degree**, **betweenness centrality**, **closeness**, and the **FR curvature** to add a variety of filtration types.
> > - We analyzed the theoretical **runtime complexities** of **Theorem 4**, **Theorem 5**, and **Proposition 8**.
> > - We clarified the difference between persistent and static descriptors.
> >
> > We hope your concerns have been satisfactorily addressed, and if so, would appreciate if you could revisit your score to reflect the same. If you have any specifc questions or concerns, we would be happy to address them. Thank you again for your feedback and help to improve the paper.

---

### Comment · Area_Chair_nxPS · 2025-08-05

Dear reviewers,

Thanks for your service to this conference! Also thanks to those of you who already engaged with the rebuttal provided by the authors. Reviewer `MD6t`, please **engage with the rebuttal** and ask any follow-up questions you may have. Please also note that every reviewer needs to provide a **mandatory acknowledgment** with a final decision, which will be hidden to the authors unless you choose to discuss it.

Thanks!

  — Your AC

---

### Decision · Program_Chairs · 2025-09-17

**Decision:**

Accept (poster)

**Comment:**

# Summary

This paper proposes a novel framework for studying topological descriptors, viz., the Euler characteristic (EC) and persistent homology (PH), on graph products under so-called color-based filtrations. As its main contributions, the paper characterizes expressivity with respect to the EC and the PH on color-based filtration; it also provides algorithms for calculating such topological descriptors for graph products.

# Strengths

- Rigorous theory for a novel application
- The work is broadly relevant for the "topology in ML" community, since it not only improves the foundation of TDA/TDL, but also provides new tool for graph ML.
- The rebuttal provided more insights into the practical applicability of the work and its integration with GNNs.

# Weaknesses

- In the current version of the manuscript, experiments are _absent_, raising concerns about maturity and practical relevance.
- The paper's writeup is quite dense, requiring a large degree of technical acumen to parse.
- In original submission, the practical aspects and relevance for the ML community was somewhat underdeveloped, but the rebuttal served to alleviate these concerns.
- The exposition of the paper could be improved

# Decision and rationale

All reviewers agreed on the relevance and quality of this work and I am happy to suggest _accepting_ it for the conference. However, this verdict is contingent on providing additional empirical experiments and strengthening the "broader appeal" of the work. While the authors provided an excellent rebuttal, some reviewers raised concerns about this submission being essentially "incomplete," insofar as no experiments were present in the original version. I concur with this assessment but I believe that a strong NeurIPS paper can also be written from the purely theoretical point of view. As such, it should not be unduly penalized. That being said, I expect the authors to use the feedback re: clarity and utility to improve their work in a revision.

# Summary of the discussion

- Reviewer `MD6t`: Leaned towards accepting the paper; citing concerns about readability and overall completeness of the work. Authors promised improvements, which partially addressed the reviewer's concerns.

- Reviewer `Ytkj`: Endorse the paper after the rebuttal, since the authors provided additional experiments and promised a clearer motivation.

- Reviewer `ETh2`: Leaned towards accepting the paper but mentioned issues with exposition; authors promised to address these concerns in a revision.

- Reviewer `6YNL`: Was concerned about the lack of experiments; raised their score after authors promised new experiments.

- Reviewer `quwK`: Appreciated the theoretical results, but cited concerns about premature submission with major results being missinga and only appearing in the rebuttal. Authors promised changes and were able to provide additional clarifications; I share these concerns and thus base my overall positive verdict on the authors' willingness to implement these changes.

The rebuttal proved to be highly useful in this case, with many reviewers engaging very strongly with the authors.